# Effect of Neutron Irradiation on the Mechanical Properties, Swelling and Creep of Austenitic Stainless Steels

**DOI:** 10.3390/ma14102622

**Published:** 2021-05-17

**Authors:** Malcolm Griffiths

**Affiliations:** 1Department Mechanical & Materials Engineering, Queens University, Kingston, ON K7L 3N6, Canada; malcolm.griffiths@queensu.ca; 2Department of Mechanical & Aerospace Engineering, Carleton University, Ottawa, ON K1S 5B6, Canada; 3ANT International, 448 50 Tollered, Sweden

**Keywords:** austenitic stainless steel, irradiation, nuclear reactors, swelling, irradiation creep, mechanical properties, microstructure, rate theory modelling, creep rupture, deformation, dislocation slip, twinning, martensite

## Abstract

Austenitic stainless steels are used for core internal structures in sodium-cooled fast reactors (SFRs) and light-water reactors (LWRs) because of their high strength and retained toughness after irradiation (up to 80 dpa in LWRs), unlike ferritic steels that are embrittled at low doses (<1 dpa). For fast reactors, operating temperatures vary from 400 to 550 °C for the internal structures and up to 650 °C for the fuel cladding. The internal structures of the LWRs operate at temperatures between approximately 270 and 320 °C although some parts can be hotter (more than 400 °C) because of localised nuclear heating. The ongoing operability relies on being able to understand and predict how the mechanical properties and dimensional stability change over extended periods of operation. Test reactor irradiations and power reactor operating experience over more than 50 years has resulted in the accumulation of a large amount of data from which one can assess the effects of irradiation on the properties of austenitic stainless steels. The effect of irradiation on the intrinsic mechanical properties (strength, ductility, toughness, etc.) and dimensional stability derived from in- and out-reactor (post-irradiation) measurements and tests will be described and discussed. The main observations will be assessed using radiation damage and gas production models. Rate theory models will be used to show how the microstructural changes during irradiation affect mechanical properties and dimensional stability.

## 1. Introduction

Austenitic stainless steels (SS) are widely used within the cores of sodium fast reactors (SFRs) and in light-water reactors (LWRs). In contrast, most of the core components in reactors operating with natural uranium fuel are made with Zr alloys to maintain a high thermal neutron economy. In the case of the Canada Deuterium Uranium (CANDU) reactor, the external containment for the moderator (the calandria vessel) is, however, made from 304 SS. The other heavy water reactor (HWR) design is that of the Atucha reactor. The Atucha reactor uses a mix of natural and enriched uranium (0.85%), with heavy water for moderation and cooling. It is like an LWR in having a single pressure vessel made of low-alloy ferritic steel. There is an inner moderator tank made from 347 SS.

For fast reactors, operating temperatures vary from 400 to 550 °C for the internal structures and up to 650 °C for the fuel cladding. The internal structures of the LWRs and HWRs operate at lower temperatures, typically between approximately 250 and 320 °C, although some parts can be hotter (more than 400 °C) because of localised nuclear heating [1]. Austenitic stainless steels are used in reactor cores because of their high strength and toughness. They retain sufficient toughness for operability after irradiation in LWRs to doses of approximately 80 dpa [1], unlike ferritic and ferritic-martensitic steels that are easily embrittled by irradiation at very low doses (<1 dpa). Ferritic steels are used for pressure vessels and for pressurized piping because of their high yield strengths but they are not used within the core region because of embrittlement at low neutron fluences.

Austenitic stainless steels have proven to be very resilient alloys for nuclear applications over many years of operation. Continued operation and/or plant life extension relies on being able to predict how the properties will evolve over extended periods. Operating experience over more than 50 years has resulted in the accumulation of data from which one can assess the irradiation response of core internals made from austenitic stainless steels that ultimately dictates the operating life of the reactors.

In this paper, the effects of irradiation on mechanical properties of austenitic stainless steels in nuclear reactors will be reviewed and assessed by radiation damage and rate theory modelling, with particular emphasis on the role of transmutation in producing He, which causes embrittlement. The focus will be limited to the effect of irradiation on intrinsic mechanical properties derived from post-irradiation, out-reactor tests. The effect of operating environment on properties such as corrosion and irradiation-assisted stress corrosion cracking (IASCC) will not be covered. Rate theory models will be used to assess swelling and irradiation creep based on microstructure evolution during irradiation.

## 2. Physical Metallurgy

The compositions of common austenitic alloys used in nuclear reactors are shown in Table 1. The most common alloys used in pressurized water reactor (PWR) and boiling water reactor (BWR) cores are the 304 and 316 stainless steels. Austenitic alloys are almost 100% face-centred-cubic (FCC) γ-phase. The Schaeffler diagram (Figure 1) shows that both 304 and 316 stainless steels retain >95% volume fraction of the γ-austenitic FCC-phase crystal structure, with a small volume fraction of ferrite, after welding [2]. The main compositional difference between 316 SS and 304 SS is the Mo addition in 316 SS, which imparts improved corrosion properties. The effect of various minor alloying or impurity elements, such as carbon, manganese, silicon and phosphorus, on properties such as swelling has been reviewed by Garner [3]. In these steels, manganese and silicon are present to aid in processing. Silicon, in particular, is important because it appears to suppress void swelling [4], without significantly affecting other properties. Alloys 286 and 316 are used for specialised components (e.g., end-fittings and filter grids) of some fuel assemblies in PWR and BWR reactors. Most large core internal structures such as the core barrel in PWRs, core shrouds in BWRs, and other core support structures, are made from 304 SS. Some reactivity mechanisms (control rods and flux thimbles) are made from 316 SS. Many bolts are made from all of the stainless steels listed in Table 1. SFRs have used 316 SS mostly for internal structures and fuel pins.

According to Masiasz and Busby [5], the steels shown in Table 1 that have added Mo (316) or are stabilised with Ti (321) or Nb (347) also have reasonably good elevated temperature strength and creep resistance.

Stainless steels are primarily composed of Fe, Cr and Ni and occupy a narrow section of the Fe–Cr–Ni phase diagram at 400 °C, outlined in green in Figure 2. The FCC austenite γ-phase is the main phase of austenitic stainless steels, as the name states, although there may be a small volume of BCC α-ferrite phase, especially after welding. Because most reactor-grade stainless steels contain between 10 and 30 wt% Ni, they have large volume fractions of austenite phase and have properties closer to that of Ni alloys rather than the low-alloy body-centred-cubic (BCC) ferritic steels. Although they do not contain high concentrations of Ni, the effect of Ni in promoting He production in austenitic stainless steels can still be important, especially when the thermal neutron flux is high.

By and large, Fe–Cr–Ni alloys are classed as austenitic steels (as opposed to austenitic Ni alloys) when the Ni content is between approximately 8 wt% and 28 wt%. Alloys are also classed as stainless steels when the Cr content is >10.5 wt%. The most widely used austenite steel in nuclear reactors is 304 SS, also known as 18/8 because it has a composition of approximately 18 wt% chromium and 8–10 wt% nickel. Austenitic steels have FCC γ-phase (austenite) as their primary phase in the fully austenitic alloys but may also contain some BCC α-phase in the duplex alloys as well as minor secondary phases, including carbides and intermetallic phases such as the tetragonal sigma phase. Most (except high-Ni A286) are not precipitation hardenable by heat treatment, yet they can be hardened by cold working and quenching. After deformation, the 304 and 316 steels may contain hexagonal-close-packed (HCP) ε-martensite, which is easily created from the shear of the parent FCC γ-phase. This HCP ε-martensite phase should not be confused with body-centred tetragonal (BCT) α’-martensite that forms from quenching of low-alloy carbon steels because carbon in solid solution distorts the BCC lattice.

Metastable martensite associated with the α-phase has a BCT crystal structure. It is formed in ferritic steels when the cooling rate from the high-temperature austenitic γ-phase phase, which would normally result in the BCC α-phase when the Ni concentration is low (<8 wt%), is at such a high rate that carbon atoms do not have time to diffuse out of the crystal structure in large enough quantities to form cementite (Fe_3_C). Some authors have reported that it is possible to obtain the BCC phase by deformation of the austenitic FCC phase in 304 SS [6,7]. Other authors have reported that the creation of the HCP ε-martensite is a precursor to the formation of a BCC α-martensite phase by deformation [8]. In the latter case, the BCC α-martensite phase cited by the authors is not the same as the BCT α’ martensite associated with quenched ferritic steels. There is some ambiguity regarding whether to refer to the BCC martensite as α or α’. Many authors simply refer to α’ martensite produced by shear even when it may be BCC or BCT, the BCT phase simply being regarded as a distorted BCC phase. The interplay between the different possible transformation phases has been described by Murr [9]; he refers to martensite but distinguishes between the BCC(α) and BCT(α’) forms. Here, we will adopt the same convention and refer to BCC martensite (formed by shear) as α-martensite to distinguish it from the well-known α’-martensite that is BCT formed by quenching, which may also be produced by shear according to Murr. It is unfortunate that many authors simply refer to martensite and fail to distinguish between BCT, BCC or HCP martensite phases.

The shear deformation structures that occur in deformed austenitic stainless steels have been well characterised by Shen et al. [6]. They have shown how the microstructure of deformed 304 SS is comprised of a complex combination of co-existing twin and martensite platelets together with regions of α’, which they refer to as the BCC variant. The twin and martensite structures are intermediary stages of the strain-induced martensitic transformation from FCC γ to BCC α or α’ phase, as shown in Figure 3.

One sub-family of steels are the so called metastable austenitic stainless steels (MASS) that make use of the properties of strain-induced α-martensite (presumably BCC) formed during cold work, e.g., AISI 301 and AISI 302 stainless steels [10]. They resemble AISI 304 austenitic stainless steels [10] but are richer in carbon (typically in the range 0.08–0.10 wt%). The main difference between type 301 and 302 stainless steels is the nickel content (6–8% and 8–10%, respectively) and chromium content (16–18% and 17–19%, respectively). They are used mostly for springs and high-strength components. The process of forming the α-martensite by deformation (presumably the BCC form) is a complex process involving the intersection of shear bands [11]. Note that Talonen and Hanninen [11] refer to the martensite as α’ but we will assume they mean the BCC form because they do not specify the crystal structure. The MASS steels such as 301 and 302 are not commonly used in nuclear reactors even as springs (probably because they are metastable) and will not be discussed further.

The formation of twins and the different martensite variants (pervasive in deformed neutron-irradiated austenitic stainless steels) is often manifested as platelets that can be mistaken for localised dislocation slip bands or “channels” in irradiated materials. The distinction between twin/martensite platelets and localised slip bands has a profound effect on how one interprets the effects of irradiation on mechanical properties.

## 3. Mechanical Properties

### 3.1. Irradiation Hardening

Irradiation with neutrons hardens the material by introducing barriers to dislocation slip in the form of point defect clusters. At low temperatures, these clusters can be vacancy stacking fault tetrahedra and small, faulted interstitial loops on the close-packed {111} planes (Frank loops). At high temperatures, the vacancy clusters tend to be cavities and the faulted Frank loops tend to unfault and form a coarser dislocation network as they become larger [3,12]. Other chemical changes can occur as a result of diffusional mass transport, resulting in precipitate formation [12,13] that also contributes to hardening. Segregation of alloying elements and impurities at sinks such as grain boundaries also occurs but that has more of an impact on corrosion or stress corrosion cracking [12,13] and will not be covered here. It is sufficient for the sake of this paper to consider the hardening effects of interstitial dislocation loops and cavities. Both act as barriers to dislocation slip and harden the material. Precipitates may also form and harden the material but they will be considered the same as other point defect clusters in the time evolution of their hardening effect. The hardening effect is a function of the barrier strength, size and spacing. The resistance to dislocation passage is determined by a strengthening factor (α) and the spacing of the different barriers to slip in the slip plane (λ). The increment in yield strength is
(1)$$\Delta \sigma_{0}=\frac{\alpha }{\lambda }=\alpha \sqrt{N\cdot d}$$
where *N* is the number density of impeding objects and *d* is their diameter [14]. The strength factor (*α*) is also a function of the barrier diameter (*d*). Tan and Busby [15] have shown the effect of cluster size on strengthening factor, Figure 4. They have shown that, as loops and cavities increase in size, the strength factor increases (as one might expect). Surprisingly, cavities have a higher strengthening factor than dislocation loops of the same diameter. This is important when there is a propensity for getting high cavity densities in cases where there is a substantial amount of He present as a result of transmutation.

### 3.2. Channelling

Localised deformation known as channelling reduces the ductility of the deformed material and, in many cases, results in strain softening when the channelling involves the sweeping up of dislocation loops by the gliding dislocations, thus removing impediments to further slip in the same volume.

It is a common misinterpretation that researchers attribute all localised deformation of irradiated austenitic alloys (that may include twinning) as channelling, or strain softening, due to dislocations sweeping up dislocation loops following the mechanism originally proposed by Foreman and Sharp [16]. One macroscopic feature that coincides with localised softening (dislocation channelling) due to the sweeping up of dislocation loops is the load drop at the upper yield point in standard uniaxial tensile test curves and is best illustrated for tensile tests of neutron-irradiated Zr [17,18], as illustrated in Figure 5 [17]. At point (a), the deformation band begins to form and is fully formed at point (b). At point (c), a second deformation band on a second slip system forms, and specimen fracture begins at point (d). Because virtually all the strain occurs in the deformation band, there is little or no deformation in the rest of the specimen gauge length. This particular example is for Zr although similar stress–strain behaviour applies to austenitic stainless steels [19,20]. The first load drop in Figure 5 corresponds with channelling/strain localisation occurring on one slip system. The second load drop corresponds with additional localisation occurring for a secondary slip system. The second system helps maintain the load line through the specimen centre of gravity by compensating for the lateral displacement caused by translation arising from the first slip system by an opposite lateral displacement from the second slip system. Channelling in Zr and Zr alloys occurs on both prism and basal planes that are 90 degrees to one another [17], but can also occur on planes inclined to both, as shown in Figure 6a. The clearing of radiation damage in sheets leaves a volume of material that is mostly free of dislocations but is characterised by an uneven edge to the channel region and the presence of residual dislocations within the channel, as shown in Figure 6b [21]. In some cases for anisotropic materials such as Zr alloys, the load drop (softening) could be associated with twinning if the crystal is rotated to a softer orientation relative to the tensile axis [21].

Channelling is associated with a load drop because the volumes cleared of dislocation loops are softer pathways for additional gliding dislocations. The process of dislocation channelling involves the interaction of edge dislocations with stacking fault tetrahedra or dislocation loops so that those defects are subsumed into the gliding dislocation network as demonstrated by molecular dynamics simulations of Diaz de la Rubia et al. [22], Voskoboynikov et al. [23], and Serra and Bacon [24]. Some interactions, e.g., those involving screw dislocations and prismatic loops, simply inhibit the passage of the dislocations and by-pass of the loop can occur if the stress is high enough [24].

Strain localisation resulting in a load drop is a common phenomenon exhibited by neutron-irradiated stainless steels, as shown in Figure 7 [25]. As with Zr alloys, channelling is often manifested as a thin band of material that is relatively free of dislocations when observed in a transmission electron microscope (TEM). In austenitic stainless steels these narrow bands are often observed at the same time as twins, twinning being another method by which a material can shear, as shown in Figure 8 [19]. There are many reports of channelling by dislocation slip in austenitic stainless steels [26,27]. Other forms of bulk shear, such as twinning and martensitic transformation, can occur at the same time. According to Lee et al. [28], strain localisation in irradiated austenitic stainless steels can be complex. They state that “…deformation bands consist of piled-up dislocations, nano-twins, stacking faults, and defect-reduced channels all tied up together.”

There have been reports of channelling that may, in fact, be cases where twin and/or martensitic shear formation has occurred giving the appearance of channelling. There are many instances where linear features, which may look like classic dislocation channels at first sight, are not dislocation channels on closer inspection. Twinning is a common feature that appears to be mistaken for channelling. Twinning, naturally, rotates the crystal so that a crystal orientation that exhibits good two-beam diffraction contrast for dislocation loops, for example, will inevitably not show the same strong contrast, or any contrast, within the twinned volume. For neutron-irradiated 304 SS, the lack of contrast and the evidence for fringes associated with bands identified as channels in Figure 9, Figure 10, Figure 11, Figure 12 and Figure 13 of Onchi et al. [29], indicates that the features identified as dislocation channels are better interpreted as twins. Likewise, for unirradiated A286 stainless steel containing a high density of γ′ precipitates, fatigue deformation resulted in bands of zero contrast within a background of strong contrast using the superlattice reflections of the precipitates, see Figure 6 of Fournier et al. [30]. In both cases, the authors did not provide evidence that dislocation slip, similar to what is shown in Figure 6b, had created the bands, even though the bands were clear of either dislocation loop contrast [29] or γ′ precipitate contrast [30]. In the absence of a statement or an analysis to establish that the features were not twins, there must be some doubt about the interpretation of the observed contrast as evidence of dislocation channelling. One can conclude that tensile deformation of austenitic stainless steels may involve some slip but the localised deformation (channelling) resulting in load drops, softening and reduced ductility may not be as pervasive as some authors believe. In some cases, the observations of features that appear to be channels cleared of dislocation loops by gliding dislocations can often best be interpreted in terms of twinning.

## 4. Inter-Granular Fracture

In the absence of stress corrosion cracking, which applies to high stresses and a corrosive environment, the other main form of inter-granular fracture of irradiated material is the result of He embrittlement. Before addressing inter-granular fracture that may, or may not, be caused by He embrittlement, it is useful to summarise the sources of He generation in a neutron irradiation environment. At low thermal neutron doses, the main source of He is from the alpha-particles produced by the (*n*, α) reaction of thermal neutrons with boron. In austenitic stainless steels boron, which is an impurity, has a concentration that is typically between 2 and 11 wt ppm. Because the main isotope that produces the He is 10 B, with an isotopic abundance of approximately 20%, the maximum amount of He that can be produced is between approximately 2 and 11 appm [He] after a thermal neutron exposure of approximately 10^21^ nm^−2^, E < 0.5 eV [31,32]. Nickel, on the other hand, which constitutes approximately 10 wt% of a 300-series austenitic stainless steel, produces significantly more He than the boron after moderate neutron doses.

For Ni, neutron absorption of the main isotope (^58^Ni) creates ^59^Ni and this isotope can have a large effect on the irradiation damage production, introducing a non-linearity to the atomic displacement damage rate as a function of time. The ^59^Ni evolves as a result of the (*n*, γ) reaction with ^58^Ni and the subsequent (*n*, α), (*n*, *p*) and (*n*, γ) reactions. The reaction cross-sections for ^59^Ni are very high, especially at thermal neutron energies, and it becomes a significant contributor to both displacement damage and gas atom (He and H) production when sufficient ^59^Ni has been produced by transmutation [33,34]. In the case of the CANDU reactor, ^59^Ni reaches a peak concentration of approximately 4% of the parent ^58^Ni isotope after approximately 5 years of operation in the reactor core [35]. The ^59^Ni enhances atomic displacements and He production through the (*n*, α) reaction. The (n, α) reaction cross-section for ^59^Ni after 5 years operation in a CANDU reactor (scaled by 4%) is shown in Figure 9 and compared with the (*n*, α) cross-sections for the main naturally occurring isotopes in austenitic stainless steels, Fe, Cr and Ni. Also shown are typical reactor spectra. It is clear that the ^59^Ni is important for enhanced He production in the high thermal neutron fluxes of power reactors because the (*n*, α) cross-section is high across a large range of neutron energies. Many materials will produce He but most, except for boron, only have significant (*n*, α) cross-sections at high neutron energies.

Even without the production of ^59^Ni, Ni can still be effective in producing He in fast reactors because the high neutron energy (*n*, α) reaction cross-sections for naturally occurring Ni results in an order of magnitude higher He production per atom compared with other major alloying elements such as Cr and Fe. The enhancement of He production due to naturally occurring Ni is also high, on a per atom basis, compared with that from minor elements such as Ti, Al and Si, which have relatively high (*n*, α) cross-sections compared to Cr and Fe. Nitrogen, on the other hand, has an order of magnitude higher helium production rate compared to naturally occurring Ni in a fast reactor environment. The ^59^Ni will become important to irradiation damage production once there are significant amounts of ^59^Ni produced from the parent ^58^Ni, e.g., when the thermal neutron fluence exceeds 10^21^ n cm^−2^ (E < 0.5 eV).

During irradiation at high temperatures, bias-driven void swelling can result in significant strengthening, or embrittlement, or both. Helium is an insoluble gas produced by (*n*, α) reactions in many engineering alloys. It promotes void swelling at high temperatures; and, at low temperatures, where vacancy mobility is low, He results in the formation of a high density of bubbles (cavities). Low-temperature He embrittlement is observed in Ni alloys when the flux of He to grain boundaries is sufficiently high [36,37,38].

At low doses, the main factor contributing to reduced ductility of austenitic stainless steels is strain localisation. Even though the load drops often observed for irradiated 316 austenitic stainless steels are coincident with reduced ductility, the fracture surfaces show ductile failure at doses ~9 dpa [20]. Inter-granular failure is more predominant for material irradiated to higher doses in a PWR following Charpy impact testing in air, as shown in Figure 10a,b [39]. The shift to predominantly inter-granular fracture can be attributed to segregation of He bubbles at grain boundaries, as shown in Figure 10c [40]. Figure 10 combines information from two separate studies on irradiated 316 SS (presumably from the same flux thimbles), irradiated to approximately the same dose (~70 dpa) in a PWR environment. The fracture is perfectly inter-granular at −196 °C and mostly inter-granular with some ridging (probably from twins or slip bands) at 30 °C. The link between cavity segregation at grain boundaries and inter-granular fracture was made by Fujimoto et al. [41]. Analysis of the same material by Edwards et al. [40,42] showed that the cavities on the boundaries were small (<2 nm diameter) and densely spaced (<10 nm spacing). The cavity segregation on grain boundaries was associated with the presence of hydrogen or helium [41], or simply helium [42].

Helium is important in stabilising cavities on grain boundaries, otherwise the vacancies diffusing to the boundaries are simply absorbed. This is illustrated in Figure 11, showing increased grain boundary coverage by cavities in 316 stainless steel with increased levels of [He] after creep rupture testing with simultaneous implantation with He [43]. The micrographs (a) and (b) show He-stabilised cavities in two different samples containing 500 appm He and 2500 appm He, respectively. The micrograph (c) shows that there is enhanced accumulation of He at boundaries perpendicular to the tensile stress. The higher grain boundary cavity coverage can be directly related to the higher He content and the lower creep rupture stress.

Cavities on grain boundaries are noticeably absent for steels irradiated by electrons and in a fast reactor when there is little He produced by transmutation, as shown in Figure 12 [44]. Zones denuded of cavities are observed adjacent to the grain boundaries in the latter case, indicating that the boundaries are net sinks for vacancy point defects. Inter-granular cracking due to cavity segregation at grain boundaries is typically observed when He is present and is therefore dependent on having cavities stabilised and segregated on grain boundaries but is not necessarily dependent on the amount of swelling.

The link between [He] and lower creep rupture stresses and times prior to inter-granular failure has been made by different authors [43,45,46,47,48]. Shroeder and Batfalsky [43], concluded that all He-implanted samples exhibited a reduction in rupture times and lower ductility for the entire stress range investigated and that all He-implanted samples showed inter-granular, brittle fracture.

Rate theory calculations [49] show that the net flux of point defects to boundaries (He atoms and vacancies) is dependent on the grain interior sink strength, as shown in Figure 13 and Figure 14. In principle smaller cavities on the boundary will lead to a higher area coverage for a given stabilised cavity volume. However, observations of He embrittlement indicate that the effect is larger for higher temperatures of irradiation [46,50]. The boundaries can receive a higher flux of point defects at higher temperatures if the sink density within the grain interior is lower at high temperatures than for lower temperatures. Modelling shows that to achieve a higher area coverage at higher temperatures there has to be either: (i) a lower sink strength in the grain interior; or (ii) He trapping at freely-migrating vacancies that have a higher steady-state concentration at lower temperatures [49]. Either or both of these factors will affect and lower the He and net vacancy flux to grain boundaries at the lower temperatures. The important result from rate theory is the trend of grain boundary area coverage with increasing dose and temperature showing a decreasing rate of coverage as a function of dose at high doses and an increase in coverage as a function of temperature [49]. The limiting factor in the evolution of the grain boundary cavity structure are the point defect fluxes from the grain interior, which are non-linear with dose.

The model developed to predict the grain boundary area coverage by He-stabilised cavities for Inconel X-750 irradiated in a CANDU reactor [49], has been applied to the swelling and grain boundary cavity coverage for the PWR flux thimble studied by Fukuya et al. [39], and Edwards et al. [40], shown in Figure 10. The cavity number density in the matrix for the flux thimble irradiated to 70 dpa at 315 °C is approximately 2 × 10^23^ m^−3^ [40]. For an assumed maximum cavity diameter in the matrix = 3 nm [40], the swelling is ~0.3%. Scaling the freely-migrating-defect (FMD) production rate to match the swelling from the cavity diameter and number density given by [40], the predicted swelling and grain boundary area coverage as a function of dpa are shown in Figure 15. The predicted area coverage for the 316 SS in a PWR (~80% at 70 dpa) is a lot higher than the area coverage measured and calculated for Inconel X-750 (~30%) for a similar dpa and operating temperature [49]. The higher coverage for 316 SS irradiated in a PWR is a direct consequence of the lower cavity density in the matrix; 1 to 2 orders of magnitude lower for the PWR compared with the CANDU case for cavities of approximately the same size. The lower grain boundary coverage corresponding with a higher matrix cavity density is a direct consequence of the lower flux of point defects (He, H, vacancies and interstitials) to the boundary because they interact with the grain interior microstructure, thus reducing the net point defect flux to the grain boundaries. The higher cavity density in the grain interior for the Inconel X-750 irradiated in a CANDU reactor is a direct consequence of the higher He generation rate; ~330 appm He per dpa for the Inconel X-750 compared with ~9 appm He per dpa for the 316 SS.

From a mechanistic standpoint, research over the last 10 years has been evolving. The first thing we need to be clear about is why Ni alloys, such as Inconel X-750, have poor fracture toughness [38], when other materials irradiated in the same CANDU environment (Zr alloys) have moderately good toughness (so long as the hydride content is not too high). The main reason why Ni alloys have poor properties is because of the large concentrations of helium that are produced by transmutation [35]. The CANDU reactor has a high thermal neutron flux and that, combined with the high (*n*, γ), (*n*, *p*) and (*n*, α) reaction cross-sections for ^58^Ni and ^59^Ni, means that Inconel X-750 core components contain up to 30,000 appm He (3 at%) by end-of-life (approximately 30 years of operation). Because He stabilises cavities on boundaries and interfaces, the resultant perforation weakens the material at these locations leading to low-energy failure when subjected to an applied load. The same principle applies to austenitic stainless steels except that the severity of He embrittlement is much less because the amount of helium generated by Ni is only 15% as much as for Inconel X-750 for a given neutron spectrum. There are two competing effects, however. On the one hand a lower He production rate in 316 SS means there are fewer He atoms available to migrate to boundaries. On the other hand, a high density of cavities in the grain interior acts as sinks for point defects, lowering the net flux of all point defects to the grain boundaries. Operation at high temperatures will also promote diffusion of point defects to grain boundaries because the grain interior sink strength decreases with increasing temperature. Boron is another source for He but is limited to just a few appm He, equal to the concentration of B in the material by weight [31]. Boron impurity levels in stainless steels are typically less than 11 wtppm B [31], so the amount of He that can be generated by a thermal neutron dose of 10^21^ n.cm^−2^, E < 0.5 eV is <11 appm.

He embrittlement is characterised by inter-granular fracture surfaces (Figure 10). Whereas it is widely accepted that cavity segregation (at grain boundaries in particular) will promote brittle, inter-granular failure, there are two differing opinions regarding what controls the ultimate failure condition. In one scenario the propensity for low-energy fracture at increasingly higher doses is assumed to be dependent on a trend to more localised plastic deformation such that the degree of slip in dislocation channels correlates with cracking severity [51]. The increasingly localised deformation is manifested as lower ductility attributed to higher localised stresses where dislocation slip bands intersect with grain boundaries. The link between slip bands and cracking is confounded by not knowing whether the localised slip promotes cracking where slip bands intersect grain boundaries or vice versa. In an alternate scenario the trend for inter-granular fracture with increasing dose is attributed to the increasing perforation of the boundaries [38]. Molecular dynamics models show that the fracture toughness of grain boundaries decreases with increasing area coverage by cavities [52], as shown in Figure 16. The rate of embrittlement then depends on the rate of increase in grain boundary perforation, which is non-linear in principle, but difficult to measure in practice. Note that the values of the fracture toughness in Figure 16 are the energy release rates as a function of crack area extension and their derivation is described Xua and Demkowicz [53].

It is reasonable to assume that the grain boundary strength is simply governed by the ligament area, i.e., the contiguous area in the plane defining the grain boundary. Mechanistically, the fracture toughness of the grain boundary is dependent on the area coverage because the cavities inhibit crack blunting [52]. A crack on a boundary will normally be blunted by the emission of dislocations from the free surface (Figure 17a), which absorbs energy, as the crack advances (Figure 17b). When the boundary is perforated by cavities the fracture toughness decreases to practically zero when the coverage is only approximately 18%. There are multiple ways to speculate on what is happening—One involving increased plasticity between the bubbles and the other the creation of a hinge. The main issue is that the existence of the bubbles allows for limited, localised, plasticity. The hinge hypothesis supposes that when the cavity coverage increases to the point that dislocation emission from the crack tip is intersected (blocked) by a cavity, the dislocation activity will be restricted (localised) to the small volume of material between the crack and the cavity (Figure 17c). The crack can then advance with minimal energy absorption in a semi-brittle manner (Figure 17d), i.e., by a mechanism that is not cleavage or simple decohesion. Cracking by decohesion or cleavage require the application of traction stresses normal to the boundary that are equivalent to the theoretical strength of the material, i.e., ~E/10 or approximately 21 GPa for pure Ni, i.e., well above the stress needed to shear the material. For the hinge hypothesis the perforation of the boundary containing a planar density of cavities will result in higher stress concentrations at the hinge points (red dots). A higher cavity density reduces the ligament area between the cavities, thus resulting in dislocation source activation at the hinge at lower applied loads (F’ < F). A higher stress concentration, with or without the presence of a crack, could enable grain boundary failure even in the elastic regime of a standard tensile test as has been observed for different Ni alloys [50].

The effect of neutron irradiation on the fracture toughness of austenitic stainless steels irradiated in fast reactors and LWRs has been reviewed by Chopra and Rao [54] and is summarised in Figure 18. The data show a rapid decrease in fracture toughness over a neutron dose range of up to 10 dpa corresponding with the change in yield strength due to the evolution of the dislocation structure (see Section 6.2.2). Citing the work of Demma et al. [55], on material irradiated under LWR conditions, they noted that the irradiated fracture toughness was orientation dependent. The fracture toughness anisotropy was attributed to the presence of stringers consisting of long, narrow particles oriented in the rolling direction. These stringers were sites for quasi-cleavage, thus aiding the crack advance. Unless they only occur at grain boundaries, cracking at stringers may be regarded as trans-granular.

According to Chopra and Rao [54], the engineering measure of fracture toughness (*K_IC_*) is concerned with the critical point where unstable crack growth occurs. The value of *K_IC_* in the ASTM standard test method for measurement of fracture toughness [56] takes into account the geometry of the crack and the specimen for a given applied stress and has units of MPa. $\sqrt{m}$. Similarly, J defines the resistance of the material to stable crack extension—How much energy is input to advance the crack per unit area. Chopra and Rao [54] show that *K* and *J* are related,
(2)$$K_{IC}=\sqrt{\frac{E}{\left(1-\nu \right)}\cdot J_{IC}}$$

## 5. Trans-Granular Fracture

Trans-granular fracture is a brittle or semi-brittle fracture mode that, as the name suggests, results in the production of intra-granular facets on fracture surfaces that are not grain boundaries. One view of channel fracture is that it is linked with localised deformation, by dislocation slip in particular [3,57,58]. However, the only link with localised deformation that has been noted is because of the coincidental observation of bands of sheared cavities in the same material exhibiting trans-granular fracture. Rather than being due to dislocations sweeping up dislocation loops, the shearing observed is typical of twinning [59], which can occur multiple times within a given volume, resulting in large shears when the material is not constrained, as shown in Figure 19.

The channel deformation that has been well characterised and is common to Zr alloys [17,60] is believed to be responsible for what has been called channel fracture in irradiated 316 and 304 stainless steel [3,57,58]. Whereas there is still ductility in the presence of channeling in the case of Zr alloys, the so-called channel fracture in the irradiated steels is associated with more severe embrittlement and intra-granular cracking [3]. Analysis of AISI 316 stainless steel material exhibiting channel fracture [58], during mechanical testing after irradiation in EBR-II (built and operated by Argonne National Laboratory at the National Reactor Testing Station in Idaho Falls, ID, USA) at 380–500 °C, showed the presence of many twinned ε-martensite platelets, similar to platelets found in Inconel X-750 [61]. The platelets in the Inconel X-750 have been characterised and, in some cases, were shown to be composed of a central twin bounded by ε-martensite. The platelets in the Inconel X-750 were also associated with trans-granular fracture [61].

In the report by Hamilton et al. [58], there is some ambiguity concerning the attribution of the fracture mechanism. On the one hand, the authors talk about fracture occurring because of localised deformation in channels similar to what is observed in Zr alloys (Figure 5 and Figure 6). On the other hand, they cite quasi-cleavage at ε-martensite platelet interfaces that is also correlated with cavity (void) formation. They speculate that the quasi-cleavage occurs at low temperatures and in thin foils due to easier ε-martensite formation in the matrix. According to Hamilton et al. the ε-martensite platelets form during room temperature mechanical testing as a precursor to the formation of BCC α-martensite due to localised depletion of Ni in the matrix caused by Ni segregation at voids that had occurred during irradiation. At the same time it is assumed that channel fracture due to localised dislocation slip also occurs but at higher temperatures and in thicker material. Hamilton et al. [58] speculate that channel fracture may be initiated when the quasi-cleavage cracks intersect grain boundaries as the crack advances into thicker material.

Hamilton et al. [58] state that “deformation twins observed in the P53 duct…appear to be ε-martensite” and imply that they formed during mechanical testing after irradiation. They also state that the twin-like platelets had the same appearance as ε-martensite phase found in unirradiated material. Rather than cavities causing a depletion of Ni, thus making it energetically favourable to form ferrite, it is possible that the platelets where “quasi-cleavage” was observed were present in the original material from the initial manufacturing process and were thus locations for cavity segregation and easier cracking during room temperature mechanical testing after irradiation [58]. Even though the material was irradiated in EBR-II, the He generation is estimated to be 0.3 appm/dpa. At the highest doses examined (approximately 50 dpa), the He concentration would then be approximately 16 appm He, perhaps sufficient to contribute to He embrittlement at high irradiation temperatures [50]. Therefore, segregation of He-stabilised cavities at the incoherent platelet interfaces may be a factor to be considered.

There are two aspects of twin/ε-martensite deformation that need to be considered. Firstly, such features are likely to be present in a 304 or 316 austenitic stainless steel that has been cold worked by, say 20%, as shown by the work of Shen et al. [6], illustrated in Figure 20. Therefore, in a typical engineering application, using a deformed alloy, it is likely that the archive material will contain these defects and they will be sites for He-bubble segregation during subsequent irradiation in a He-producing nuclear environment. Secondly, post-irradiation testing to high strains (>15%) will likely also produce twins and ε-martensite. The two can be distinguished if the pre-existing platelets contain irradiation damage clusters (cavities) after irradiation that are not sheared by subsequent testing [61]. Newly created twins or ε-martensite platelets will shear any cavities within the microstructure as shown in Figure 19.

Observations of cracking at platelet inhomogeneities have been cited by Rowcliffe et al. [50], in neutron-irradiated Ni alloys. They noted that brittle failures (in the elastic regime of a tensile test) in irradiated Ni alloys (PE16 and Inconel 706) were associated with cracking along η-phase platelet interfaces. In Ni alloys, the η-phase is a Ti-rich (Ni_3_Ti) HCP phase. Such platelets would presumably have the same effect as the ε-martensite platelets reported by Hamilton et al. [58]. The platelet phase, which may contain a mixture of twin and ε-martensite, will naturally act as an inhomogeneity and platelets containing η-phase or ε-martensite would be expected to have similar effects on cracking.

## 6. Swelling

Although most empirical data on swelling are reported in terms of fast neutron fluence (E > 0.1 MeV or E > 1 MeV), some data are reported in displacements per atom (DPA). These units are employed rather than neutron fluence as a measure of dose because of the spectral differences between different reactors [35], but also because dpa can be more directly correlated with changes in physical properties. Most modelling involves a determination of that fraction of displaced atoms that migrate to sinks and affect the material dimensions and mechanical properties. The mobile point defects that affect the material properties are the freely-migrating defects (FMDs). These are the mobile clusters and point defects remaining after cascade collapse that are free to migrate to sinks such as dislocations or cavities. Whereas point defects are known to be mobile, and one can calculate their migration rate to different sinks, little is known about the properties of mobile clusters. Most calculations simply assume that all mobile radiation damage is in the form of point defects, i.e., self-interstitial atoms and vacancies [62]. Models based on point defect diffusion, including that of gaseous atoms that assist with stabilisation of large three-dimensional vacancy clusters (cavities), thus depend on knowing the gas production rate (primarily He) and the percentage of displaced atoms and associated vacancies that are mobile, the so-called freely-migrating point defects (FMDs).

### 6.1. Freely-Migrating Point Defects

The number of primary displacements caused by neutron irradiation are important when it comes to ballistic processes such as disordering and dissolution [63]. However, for processes that depend on the diffusion of point defects one needs to not only calculate the number of displaced atoms [35], but also what fraction of the displaced atoms survive spontaneous recombination in the collision cascade [64]. The number of freely-migrating point defects (FMDs) remaining after the initial collision cascade can be calculated based on primary knock-on atom (PKA) energies following the formula given by Gao et al. [65]. FMDs are responsible for cavity growth [14], and are thus important in any assessment of likely swelling, as well as for calculating irradiation creep.

Collision cascades produce a high density of point defects that can cluster and spontaneously recombine as the displacement spike cools. Although clusters formed in the cascade can contribute to hardening, processes such as irradiation-induce creep, swelling and micro-chemical segregation depend on the FMDs. The FMDs can themselves recombine and this additional mutual recombination depends on the availability of both neutral and biased sinks for the point defects within the microstructure as well as the point defect creation rate and the irradiation temperature. Typically, FMDs left over from the collapse of collision cascades created by neutrons constitute a small fraction (<10%) of the total number of atoms displaced within the cascades. However, in cases where the neutron flux is low and the gamma photon flux is high, it is possible that radiation damage from gamma photons can constitute a significant fraction of the total FMD production [31]. Most test data on austenitic stainless steels have been obtained from fast reactor irradiations at high damage rates where the contribution from gamma photons is assumed to be negligible. It will be assumed that the gamma contribution is negligible in the rate theory calculations that follow where the FMD rates have been determined only for the neutron spectrum corresponding with site 8D5 in EBR-II.

The production efficiency of FMDs from atomic displacements by neutrons is a function of the primary knock-on atom (PKA) spectrum. For EBR-II 8D5, the PKA spectrum was obtained using the SPECTER code [32]. Using the formula developed by Gao et al. [65], the effect of the PKA energy spectrum on the production of FMDs is shown for Fe atoms in Figure 21. The plots show the PKA spectrum generated by the SPECTER code [32] and the damage energy parameter, TDAM, which is the value of the energy available to create atomic displacements as defined by Norgett et al. [66]. The dpa is determined by applying TDAM for each PKA. The FMD for neutrons is then obtained by applying the cascade efficiency [65] to the displaced atoms produced for each PKA energy.

Similar data were generated for the other main alloying elements in 304 SS and 316 SS and the results are tabulated in Table 2. For each alloy, with a given composition, the contribution to DPA and FMD are tabulated. The DPA and FMD production rates are similar for both 304 SS and 316 SS. The FMD fraction for EBR-II is approximately 0.062.

When considering atomic displacements and associated FMD production in the context of the production of ^59^Ni, there is an added complication because, for high-Ni alloys, approximately half of the total atomic displacements can come from the ^59^Ni (*n*, α) reaction once sufficient ^59^Ni has been produced in a reactor with a high thermal neutron flux (such as a power reactor). The FMD fraction for most of the additional atomic displacements is dictated by the recoil energy for the (*n*, α) reaction, which is 340 keV for the reaction with thermal neutrons, yielding an FMD fraction of 0.0121 for the displacements from that recoil. The recoil energy increases with neutron energy [33], but the FMD fraction can only be less [65]. For the fast reactor spectrum shown in Figure 9, and for austenitic stainless steels that have low Ni concentrations, the added contribution from ^59^Ni to the dpa is <0.01% up to 100 dpa and is therefore negligible. The contribution to [He] is more significant with the [He]/dpa rising from 0.24 at 1 dpa to 0.255 at 100 dpa when the [^59^Ni] is ~0.25 at%. For 316 SS irradiated in EBR-II there is then a 6% increase in He/dpa due to ^59^Ni production at 100 dpa.

### 6.2. Empirical Data

When considering processes such as swelling, one can use empirical data to relate swelling to dose in terms of fast neutron fluence or dpa. The swelling itself is caused by the evolution of cavities or voids within the microstructure, the excess atoms from the voids adding to the total volume of the material. Whereas it is common for some researchers to only refer to voids [3], a distinction should be made between voids, which are volumes devoid of any atoms within a material, and gas bubbles that are similar to voids but contain gas atoms that help stabilise the voids against collapse to form vacancy platelets (prismatic loops). At a given temperature, an equilibrium bubble is one where the driving force for bubble shrinkage by vacancy emission (reducing the surface energy) is balanced by the resistance to create the vacancy (vacancy formation energy), which would shrink the cavity, and the resistance to the associated volume reduction due to the work done by the internal pressure, which is proportional to the He atom to vacancy ratio (He/V) within the cavity. During irradiation there may be an excess flux of vacancies to bubbles or voids because of the elastic attraction of interstitials with dislocation sinks and there is then a radiation-induced, non-equilibrium, condition that induces more swelling than would be expected from the presence of gaseous atoms alone. Whereas void and equilibrium bubble have specific definitions, both can be described as a cavity. Cavity is a more appropriate term when the volume is neither a void nor an equilibrium bubble. One can calculate the swelling using rate theory by accounting for net point defect fluxes (self-interstitials, vacancies, and gas atoms) to the cavities.

#### 6.2.1. Assessment Swelling Based on Empirical Data

Data on swelling in austenitic alloys such as 316 SS and Ni alloys covering temperatures up to 650 °C have been reviewed and summarised by Garner [3]. Garner has shown how swelling is affected by temperature, damage rate and also by minor alloying elements and impurities. Here, we will only consider the main alloying elements that constitute a given austenitic stainless steel.

Garner shows that, for cold-worked 316 stainless steel, the swelling rate at high fluence after irradiation in EBR-II tends to 1% per dpa at approximately 500 °C, which is approximately the peak swelling temperature, as shown in Figure 22. The swelling at 650 °C is lower, approaching that of 0.2% per dpa, and close to the swelling rate at 400 °C.

There is a relationship between swelling and Ni content with a minimum in swelling being observed for Ni concentrations between approximately 30 and 50 at % Ni, as shown in Figure 23. The reason for the reduction in swelling up to approximately 40% Ni is open to speculation and is largely not understood [3,38]. This reduction in swelling for Ni concentrations up to approximately 40 wt% is observed for both self-ion irradiation and for neutron irradiation. There is an increase in swelling as the Ni concentration increases for value >40% but for the neutron-irradiated material only. This may be the result of the effect of increased He production with increasing Ni concentration, whether due to the production of ^59^Ni or simply because the (*n*, α) cross-section is higher for naturally occurring Ni compared with other elements (Figure 9). For the EBR-II spectrum shown in Figure 9 the [He] appm/dpa production rate per dpa is 1.16 for naturally occurring Ni and increases to 1.34 when considering the effect of ^59^Ni at 100 dpa. For comparison, the [He] appm/dpa production rate per dpa for Fe is 0.14 and for Cr is 0.12, an order of magnitude lower.

One can see from Figure 22 that the change in swelling rate with dose is non-linear. There are two possible reasons for this non-linearity in swelling with increasing dose: (i) the microstructure is evolving and the optimum condition for swelling is not achieved until the cavity sink strength is comparable with the dislocation sink strength (see Section 6.2.2); (ii) there are transmutation reactions creating He that increases in concentration with increasing dose and thus contributes to an increasing cavity density and swelling rate (see Section 6.2.3). According to Garner, there are approximately 5 dpa per 10^21^ n.cm^−2^ (E > 0.1 MeV) corresponding to the data plotted in Figure 22. In EBR-II, corresponding with Figure 22, the rate of He production is approximately 0.3 appm He per primary dpa and the dpa rate is approximately 4 × 10^−7^ dpa s^−1^, with a corresponding FMD rate of approximately 2.5 × 10^−8^ dpa·s^−1^, assuming 6% FMD production efficiency. At a neutron fluence of 20 × 10^22^ n·cm^−2^ (E > 0.1 MeV), corresponding with 100 dpa, there will only be approximately 30 appm He generated.

It is instructive to model swelling when the cavities are treated as voids (no He present) and also when they contain, and accumulate, He during irradiation.

#### 6.2.2. Swelling Due to Voids (No Helium) Based on Rate Theory

The dimensional stability (resistance to creep and swelling) of stainless steels during irradiation is sensitive to the irradiation temperature. Rate theory can be used to explore the temperature dependencies of creep and swelling to gain a better understanding and improve the predictability for dimensional stability over extended operating times. If swelling occurs in the absence of He, i.e., the cavities are empty volumes that are devoid of gas, also known as voids, one can perform simple calculations to explore how biased-driven swelling is a function of the evolving microstructure.

Contrary to Garner’s claim that swelling achieves a steady-state condition of 1%/dpa, rate theory shows us that the swelling rate cannot be constant if the microstructure is changing. The swelling rate should first increase and then decrease with increasing dose as the density (sink strength) of the voids evolves from a low value and then overtakes the sink strength of the dislocations, i.e., a constant steady-state swelling rate as a function of swelling simply based on biased-driven swelling is highly unlikely.

The number density and/or size of voids increases monotonically with increasing dose [3]. For a material containing a high dislocation density (from cold working or dislocation loop formation), as the void number densities and sizes increase the swelling rate will also increase up to a point where the void sink strength is comparable with the dislocation sink strength [38]. At this point the swelling rate will have reached its maximum value.

If the cavity size or density (sink strength) continues to increase, the swelling rate will decrease and approach zero as the numbers of interstitial point defects annihilating at voids becomes comparable with the annihilation rate of the vacancies at the same sinks. When interstitial and vacancy point defects annihilate at the same type of sink there is effective recombination and there will be no further contribution to the swelling. The effective recombination rate of vacancy and interstitial point defects increases with the void sink strength, and the swelling rate must eventually decrease to zero when the neutral sink strength (that due to voids) increases to a point where all the interstitial and vacancy capture is dominated by the same sink, the voids.

Modelling the evolution of the void structure is complicated and subject to large uncertainties in the values of the parameters. The sink evolution can be assessed from experimental data and one can thus project how the swelling rate will change as the void structure evolves. Swelling is largely controlled by the migration of freely-migrating point defects. At temperatures where vacancy point defects are mobile one can calculate the swelling rate using rate theory for a give microstructure, temperature, and production rate of FMDs.

The dislocation structure is a combination of the original network dislocations introduced by fabrication and dislocation loops caused by interstitial point defect clustering. The network dislocations can recover and new dislocation loops can be created by point defect clustering and growth depending on the temperature and damage rate. Empirically, the dislocation structure evolves to a more-or-less constant state even though the voids may continue evolving. The swelling can then be calculated assuming a constant dislocation structure and a void structure that is evolving at a given rate.

A simple example of a rate theory formulation to describe swelling is shown in Equations (3) and (4), where the swelling is assumed to be dictated by the net biased flow of interstitials to network dislocations and dislocation loops formed from radiation damage. It is assumed that mutual recombination of vacancy and interstitial point defects is negligible, i.e., we are in a sink-dominated regime. The swelling is dictated by the bias factor (b) that effectively increases the sink strength of a given sink for a particular point defect; in this case the bias of dislocations for attracting interstitial point defects [67]. In Equations (3) and (4), the unbiased dislocation sink strength is given by the dislocation density, ρ_d_, in units of m^−2^, and the unbiased void sink strength denoted by ρ_c_, i.e., 4πrN, also in units of m^−2^, where r is the void radius in m and N is the number density in m^−3^ [67]. The subscripts in these parameters refer to dislocations (d) and cavities/voids (c). The bias parameter (*b*) represents the increased probability that interstitial point defects will be absorbed at a dislocation sink [67,68]. Representing the flux of point defects (atom fraction per second) to various sinks (s) by J_s_, where the subscript s is c for cavity/void sinks and d for dislocation sinks, the fluxes to sinks resulting in volume expansion are given by: (3)$$J_{d}=[\frac{\left(1+b\right)\cdot \rho d\;}{\left(1+b\right)\cdot \rho_{d}+\rho_{c}}-\frac{\rho_{d}\;}{\rho_{d}+\rho_{c}}]\cdot \Phi$$
(4)$$J_{c}=[\frac{\rho_{c}\;}{\left(1+b\right)\cdot \rho_{d}+\rho_{c}}-\frac{\rho_{c}\;}{\rho_{d}+\rho_{c}}]\cdot \Phi$$
where ρ_d_ and ρ_c_ are the unbiased sink strengths for dislocations and cavities/voids and *Φ* is the atomic displacement damage rate in dpa s^−1^. To distinguish between interstitials and vacancies, interstitials are defined as positive and vacancies as negative in terms of the point defect flux; an interstitial adding and a vacancy subtracting, one atom at a given sink type. In order to relate these changes to a strain (either a linear increase in dimensions or a change in volume) the units are given in atom fraction. For a dose of 1 dpa every atom in the material has been displaced once. For 5% cascade damage efficiency the FMD fraction for each displaced atom is 0.05. If there was an excess flow of only 10% of the FMDs that annihilated at their respective sinks (vacancies at voids and interstitials at dislocations), the swelling would then be 0.5% per dpa.

An illustration of the dependence of swelling rate on dislocation and void sink strengths for an assumed set of point defect properties is shown in Figure 24. The calculated swelling rates as a function of an evolving void density (assume 2 nm diameter voids) for two nominal dislocation densities corresponding with unirradiated (2 × 10^14^ m^−2^) and irradiated (2 × 10^15^ m^−2^) materials are shown in Figure 25 [38]. In this particular example, the nominal cascade efficiency has been assumed to be 1%. The plot shows that as the void density evolves the swelling rate passes through a peak when the net partitioning of point defects to the different sink types is optimal. This occurs when the ratio of the unbiased vacancy and interstitial sink strengths is equal to $\sqrt{\left(1+b\right)}$.

For more complicated sink structures, and including the effect of He, a numerical solution is the easiest way to assess the peak conditions for swelling. If He is being continuously generated, the He itself has an effect on the propensity for vacancies to annihilate at cavities; the He increases the capacity of the cavities to absorb more vacancies.

#### 6.2.3. Swelling Based on Rate Theory (with Helium)

Following the work of Mansur et al. [69], the growth rate of He-containing cavities (radius $r_{c}$) is a function of the net point defect fluxes to the cavity, the He content that controls the cavity pressure, the surface energy and the vacancy formation energy so that,
(5)$$\frac{\mathrm{dr}}{\mathrm{dt}}=\frac{\Omega }{r_{c}}\cdot \left[D_{i}C_{i}-D_{v}C_{v}-D_{v}C_{v}^{\mathrm{th}}\right]$$
where
(6)$$C_{v}^{th}=exp\left(-E_{f}^{v}-P_{c}\cdot \Omega +\frac{2\cdot \gamma \cdot \Omega }{r_{c}}\right)\cdot \frac{1}{k\cdot T}$$
$C_{v}^{th}$ is the vacancy thermal equilibrium concentration at the surface of the cavity, $E_{f}^{v}$ is the vacancy formation energy, $P_{c}$ is the internal pressure due to He [70], Ω is the atomic volume, $r_{c}$ is the cavity radius. The flux of He atoms to cavities that dictates the value of $P_{c}$ is calculated by treating the He atoms as unbiased diffusing species so that their accumulation at a given sink is simply a function of the relative unbiased sink strengths. The He diffusion parameters used are those outlined by Philipps et al. [71]. The interstitial and vacancy fluxes are determined according to Equations (7)–(11) [67].
(7)$$\phi -\underset{s}{\sum }k_{v}^{2}\;D_{v}C_{v}-aC_{i}C_{v}=0$$
(8)$$\phi -\underset{s}{\sum }k_{i}^{2}\;D_{i}C_{i}-aC_{i}C_{v}=0$$
(9)$$k_{i,v}^{2}\;D_{i,v}C_{i,v}=\phi \;F\left(\eta \right)$$
(10)$$F\left(\eta \right)=\frac{2}{\eta }\left\{{\left(1+\eta \right)}^{0.5}-1\right\}$$
(11)$$\eta =\frac{4\;a\;\phi }{D_{v}\;D_{i}\;k_{v}^{2}\;k_{i}^{2}}$$
where ϕ is the freely-migrating point defect generation rate in atom fraction per second, ($k_{i,v}^{2}$) is the sink strength summed over all sinks (s) and has units of m^−2^, ($D_{i,v}$) is the diffusion coefficient and has units of m^2^ s^−1^, ($C_{i,v}$) is the atom fractional point defect concentration in the diffusive medium and (a) is a recombination rate parameter, which varies between approximately 10^20^$\cdot D_{i}$ [67] and 100 × 10^20^$\cdot D_{i}$ [72]. The point defects are designated (*i*) for interstitial atoms, and (*v*) for vacancies. The rate of flow of point defects to a given sink is determined by the concentration difference in the medium midway between the sinks $\left(C_{\alpha }\right)$ and the concentrations at the sink surfaces $\left(C_{\alpha }^{th}\right)$.

Using the Carnahan–Starling equation of state to calculate the cavity pressure [70], the cavity growth rates as a function of cavity diameter are then determined from Equations (5) and (6) taking into account the accumulation of He that is being constantly generated. Maintaining a constant cavity number density (determined by nucleation in the early stages of irradiation), a calculated cavity growth rate can be computed to investigate the effect of temperature as it relates to likely swelling to best match the data shown in Figure 22. Apart from assuming a fixed cavity density in the model, which is a variable, a surface energy of 2 J m^−2^ and known models for cavity pressures [70], one can calculate the likely swelling using the FMD and He generation rates corresponding with EBR-II 8D5.

To model the swelling in Figure 22, the swelling is computed assuming a fixed cavity number density (as if the cavity nucleation occurred early in the irradiation and remained fixed thereafter). The swelling is then derived from the growth of the cavities by applying the fluxes of point defects and He atoms and applying Equations (5)–(11). The mean cavity diameters calculated from the rate theory vary according to the irradiation temperature and the swelling is computed accordingly. Using yield stress data from Garnier et al. [73], the dislocation density evolution can be modelled as shown in Figure 26. Applying recombination coefficients of 10^20^∙D_i_ [67] or 100 × 10^20^∙D_i_ [72], the swelling can be calculated assuming a fixed nucleation density for the cavities. The swelling is then determined from the cavity growth with no change in number density. The calculated swelling for 316 SS at different temperatures for a nucleation density of 10^20^ m^−3^, which is at the low end of the observed range [74], are shown in Figure 27. The best agreement with the peak swelling rate consistent with data from Garner [3], which exhibit a peak swelling rate at approximately 500 °C, is with a recombination coefficient of 10^20^∙D_i_ [67]. However, the best agreement at temperatures above and below the peak swelling temperature arises from using a recombination coefficient of 100 × 10^20^∙D_i_ [72]. The magnitude of the calculated swelling is low in comparison and the model output can be better matched to the data using a cavity nucleation density of 10^21^ m^−3^, which is at the high end of the observed range [74], as shown in Figure 28. Whereas the response for the peak swelling temperature (approximately 500 °C) is better for the smaller recombination coefficient (Figure 28a), the match with the experimental data at lower temperatures is better using the larger recombination coefficient (Figure 28b). Neither one matches the experimental data at temperatures >600 °C. One can conclude, with a limited number of trials, that the best fit to the peak swelling data can be achieved using a recombination coefficient of 10^20^∙D_i_ [67], and a cavity number density of 10^21^ m^−3^ [74].

## 7. Irradiation Creep

Irradiation creep is often described in terms of a primary, secondary and tertiary stages. The primary stage, if it exists, can be considered as an initial increase in strain that is apparent from the first measurement in an irradiation creep test, generally no less than a few hundred hours [68,75]. The secondary stage is often considered to represent a constant (steady-state) creep rate (for a given flux and temperature). The reality is that the secondary creep behaviour evolves continuously as the microstructure evolves. In the early stages of irradiation (<10 dpa) the microstructure evolution primarily involves the dislocation structure (increasing density of dislocation loops or decreasing network dislocation density due to recovery and/or dislocation creep). At high doses (>10 dpa), the creep evolution primarily involves changes in the cavity structure (swelling) for a constant dislocation structure. Tertiary creep is essentially the acceleration of creep because of increasing true stress (in a necked region) as the specimen geometry changes due to the creep strain—the elevated stresses result in accelerating creep rates until failure occurs (creep rupture).

The history of the development of irradiation creep concepts that apply to austenitic stainless steels until 1983 has been described and discussed in detail by Franklin et al. [76], and Matthew and Finnis [77]. At the time of their reviews, the focus was on mechanisms that relied on the effect of stress on dislocation loop nucleation (SIPN), diffusion to dislocations (SIPA) and the enhancement of climb of gliding dislocations over obstacles (SICG). In more recent reviews of irradiation creep in austenitic stainless steels by Garner [3,78], the role of swelling was a strong focus. Swelling (when present) is deemed an important factor in the dimensional changes observed during irradiation. This is due to the direct impact on the volume change of the material and because the voids created under irradiation, associated with swelling, increase the density of sinks for vacancy point defects (neutral sinks) and this results in a higher flux of interstitials to network dislocations and dislocation loops, thus inducing an enhanced irradiation creep rate.

At low stresses (<120 MPa in PWRs), stainless steel components exhibit irradiation creep that is enhanced by irradiation and is linear with stress. In fast reactors at high temperatures, above 400 °C, when the swelling is significant, the steady-state irradiation creep can be described by a simple empirical law [79], based on work by Foster et al. [80],
(12)$$\dot{\bar{\varepsilon }}=(B_{0}+D\dot{S})\bar{\sigma }$$
where $B_{0}$ is the steady-state irradiation creep rate compliance in the absence of swelling in units of dpa^−1^ MPa^−1^, D is the irradiation creep-swelling compliance in units of MPa^−1^ and $\dot{S}$ is the linear swelling rate in units of dpa^−1^. The compliance ($B_{0}+D\dot{S})$ relates the equivalent strain $\dot{\bar{\varepsilon }}$ to the equivalent stress ($\bar{\sigma })$ [79] as defined by,
(13)$$\bar{\sigma }=\sqrt{{\left(\sigma_{1}-\sigma_{2}\right)}^{2}+{\left(\sigma_{2}-\sigma_{3}\right)}^{2}+{\left(\sigma_{3}-\sigma_{1}\right)}^{2}}$$
(14)$$\dot{\bar{\varepsilon }}=\sqrt{{\left({\dot{\varepsilon }}_{1}-{\dot{\varepsilon }}_{2}\right)}^{2}+{\left({\dot{\varepsilon }}_{2}-{\dot{\varepsilon }}_{3}\right)}^{2}+{\left({\dot{\varepsilon }}_{3}-{\dot{\varepsilon }}_{1}\right)}^{2}}$$
where $\sigma_{d}$ and $\dot{\varepsilon_{d}}$ are the stress and strain rates in each of three orthogonal directions (d) referred to the principal axes of stress. When the response is isotropic, use of these expressions allows one to compare results from different types of tests with different stress states, e.g., uniaxial and biaxial. This simple relationship cannot be applied when there is any degree of anisotropic response of the material induced by the thermo-mechanical processing [81]. For an anisotropic material, whereas the calculation of the effective stress is the same, the expression for effective strain is complicated and depends on the anisotropy factors as well as the stress state [82].

The expressions for effective stress and strain are based on the Von-Mises yield criterion. Hill derived a yield criterion equation for anisotropic materials [83]. Hill’s yield criterion is given by,
(15)$$G{\left(\sigma_{1}-\sigma_{2}\right)}^{2}+F{\left(\sigma_{2}-\sigma_{3}\right)}^{2}+H{\left(\sigma_{3}-\sigma_{1}\right)}^{2}=k$$
and is also the representation quadric for the tensor relating strain rate to stress. In terms of a tensor equation for creep strain rate the formal definition of the compliance (referred to the principal axes of stress) is
(16)$$\left(\begin{array}{c} \dot{\varepsilon_{1}}\\ \dot{\varepsilon_{2}}\\ \dot{\varepsilon_{3}} \end{array}\right)=f\left(M,\Phi ,T\right)\cdot \left(\begin{array}{ccc} G+H & -G & -H\\ -G & G+F & -F\\ -H & -F & H+F \end{array}\right)\cdot \left(\begin{array}{c} \sigma_{1}\\ \sigma_{2}\\ \sigma_{3} \end{array}\right)$$
where f(M,Φ,T) is the magnitude of the creep compliance relating the magnitude of the strain rate to the stress state $\left(\sigma_{1}\;\sigma_{2}\;\sigma_{3}\right)$, referred to principal axes, for a given temperature (T), neutron flux (Φ) and material or microstructure (*M*). The equivalence of the compliance tensor represented in this manner with Hill’s yield equation can be demonstrated by determining the characteristic equation for the compliance tensor given by,
(17)$$\sigma^{T}K\sigma =\left(\sigma_{1}\;\sigma_{2}\;\sigma_{3}\right)\cdot \left(\begin{array}{ccc} G+H & -G & -H\\ -G & G+F & -F\\ -H & -F & H+F \end{array}\right)\cdot \left(\begin{array}{c} \sigma_{1}\\ \sigma_{2}\\ \sigma_{3} \end{array}\right)$$
giving
(18)$$\sigma^{T}K\sigma =G{\left(\sigma_{1}-\sigma_{2}\right)}^{2}+\;F{\left(\sigma_{2}-\sigma_{3}\right)}^{2}+\;H{\left(\sigma_{3}-\sigma_{1}\right)}^{2}$$
one can determine *F*, *G* and *H* relative to the component coordinate system by testing and measuring the creep strain rate of tensile specimens along each of the three axes, which gives *G* + *H*, *G* + *F* and *H* + *F*. This is often not possible to do for a component such as a thin-walled tube such as a flux thimble or fuel pin. One can obtain enough information to determine *F*, *G* and *H* from two tensile specimens (perhaps even bent beam stress relaxation specimens) if one can simultaneously measure the strain rate in both the long and short axes of at least one specimen [75]. Performing tests to obtain *F*, *G* and *H* relevant to the component of interest is difficult in an irradiation environment.

Most assessments of irradiation creep in austenitic stainless steels intrinsically assume that the material response is isotropic, which may, or may not, be valid. For the isotropic case [79], *F* = *G* = *H* and one can choose their absolute values depending on the compliance f(M,Φ,T), which equals $B_{0}+D\dot{S}$, or $\dot{\bar{\varepsilon }}/\bar{\sigma }$, when *F* = *G* = *H* = 1/3. As one might expect, only for the isotropic case can $B_{0}$ and D be determined experimentally based on the expressions for equivalent stress and strain given by Equations (13) and (14). 

In applying Equation (12), there is some ambiguity whether the swelling rate ($\dot{S})$ is that measured with or (separately) without stress. The effect of stress on swelling was not known and subject to speculation at the time the equation was formulated [79]. The effect of temperature on irradiation creep is assumed to be partly captured by the effect of temperature on swelling [3]. The physics behind the two empirical terms, B_0_ and D, including the temperature dependence, has been reviewed and discussed by Was and Ukai [12] and Onimus et al. [68]. The swelling-independent term (B_0_) has been shown to have a negative temperature dependence [84] and other work has shown an insensitivity to temperature [46] for temperatures > 300 °C. However, Grossbeck and Mansur [85] also showed that the irradiation creep rate increased at low temperature (~60 °C), consistent with findings from stress relaxation tests [86].

Radiation both enhances creep, by promoting climb of dislocations over obstacles and through simple mass transport, but also suppresses creep from dislocation glide because of point defect clustering. Whether dislocation creep is enhanced or suppressed by irradiation depends on the combined effect of these two processes. The negative temperature dependence for irradiation creep at low temperatures is common to other austenitic alloys (Ni alloys) and has been explained based on enhanced recombination for temperatures < 300 °C in materials with vacancy migration energies of approximately 1.4 eV [37,38]. For damage rates typical of power reactors, the increase in mutual recombination at low irradiation temperatures results in less damage and less hardening for a given dpa compared with irradiation temperatures ≥ 300 °C, thus allowing dislocations to glide more freely. This implies that at low enough temperatures the most important factor dictating the creep enhancement is the absence of hardening centres to inhibit dislocation glide.

### 7.1. Swelling-Independent Irradiation Creep (B_0_)

For creep at high temperatures, it seems that recovery of the dislocation network structure is significant and lower creep at higher temperatures is the result of lower dislocation densities in stainless steels [78,87,88]. At lower temperatures, when recovery of any cold-worked dislocation structure is not apparent, there is often suppression of dislocation creep because of radiation damage clustering [68,75] mainly from dislocation loop formation.

It seems that some form of SICG mechanism is probably dominant when it comes to explaining the B_0_ term. The effect of hardening in reducing irradiation creep by glide is illustrated from an analysis of the 316 SS data of Lewthwaite and Mosedale [89]. Creep measurements from spring relaxation at low doses were obtained for various stresses and for a range of neutron fluxes. The strain as a function of dpa is plotted in Figure 29a. Taking the slope and dividing by stress gives the creep compliance (B_0_) at each stage of the creep evolution. Plotting the compliance against dose shows that the creep compliance (B_0_) decreases as the dose increases and reaches a steady-state (constant) condition after approximately 5–10 dpa. The steady-state compliance for the data of Lewthwaite and Mosedale is approximately 3 × 10^−6^ MPa^−1^ dpa^−1^ at doses > 5 dpa, as shown in Figure 29b. For most materials B_0_ varies from 0.5 to 4 × 10^−6^ MPa^−1^ dpa^−1^ [12,68].

According to Garner [78], part of the measured variation in B_0_ is attributed to the difficulty in separating the swelling strain and the irradiation creep strain. That part of the steady-state irradiation creep that comes from dislocation slip, which may also be enhanced by diffusional mass transport via the stress-induced climb and glide (SICG) mechanism, may be considered separate from the swelling-dependent part and is thus represented by B_0_ [68].

The saturation in B_0_ at 5–10 dpa (Figure 29b) is consistent with the evolution of the radiation damage dislocation loop structure that hardens the material and suppresses dislocation glide as evidenced by yield strength measurements in 316 SS (see Figure 26). Although some degree of climb will inevitably contribute to the steady-state creep behaviour at doses >5–10 dpa, the reduction rather than augmentation of the irradiation creep rate with increasing dose implies that dislocation glide is the dominant creep mechanism up to 5 dpa. How much of the ensuing creep can be attributed to SICG and how much to simple mass transport cannot be discerned without more data. However, the fact that there is, at the same time, a somewhat contradictory insensitivity to the amount of cold work in the “steady state”, as discussed by Onimus et al. [68], leads one to hypothesis that there are two factors at play concerning the role of network dislocations in being sources for strain through: (i) glide and (ii) climb. Irradiation can both enhance creep by creating new sinks for point defects in the form of dislocation loops at low doses and cavities at high doses; there is also a suppressing effect of these defect clusters by inhibiting glide. From the perspective of diffusional mass transport (climb), it is the ratio of the sinks rather than their absolute magnitude that is important. In this context, a change in the ratio of the sinks for different point defects is important (see Figure 24 and Figure 25).

### 7.2. Swelling-Dependent Irradiation Creep (D)

The D term is, by definition, a factor used to link the irradiation creep rate to the swelling rate. It is difficult to ascertain how much of the dependence of irradiation creep on swelling is simply because of the linear strain from swelling and how much is because of a true enhancement of the creep, i.e., whether D is effectively a multiplier of whatever the linear swelling would have been in the absence of a stress. It is difficult to interpret Equation (12) when the swelling and the creep rate are inter-linked, which is what rate theory would indicate. Some understanding of the physics can be obtained using a simple rate theory model.

Anisotropic swelling [90], and any non-linear variation of swelling with stress, complicates the analysis of swelling and irradiation creep strains [91,92,93]. Part of the complication in deriving an empirical expression that includes swelling and creep is the assumption that swelling and creep are independent and separable. Because swelling is dependent on diffusional mass transport it must also be inter-related with that part of the creep that is also dependent on diffusional mass transport. If stress has any effect on the diffusion of point defects, the swelling itself cannot be independent of stress.

An easier way to understand the effect of stress on mass transport is to consider the strains in three orthogonal directions for a material subject to a simple uniaxial stress. The strain from the interstitial atoms that represent the mass removed from the cavities will be partitioned in the three directions according to the applied stress assuming that the stress induces a bias to the interstitial flux in the appropriate direction. A simple rate theory construct is shown schematically for the case of a uniaxial stress in Figure 30. It is assumed that the kinetics are sink dominated (temperatures > 300 °C) and recombination can therefore be ignored. The normal dislocation bias (b), calculated following the expressions given by Heald and Speight [67], is taken as 0.3. Following Woo [94], it is further assumed that the stress induces an additional interstitial bias (s) that is anisotropic and (we will assume) half the value of the normal dislocation bias, i.e., 0.15, for a tress of 100 MPa. The effect of this stress-induced bias can be explored in the context of the resultant creep strain and what it means in terms of D. It is assumed that there are only two types of sink: cavities and dislocations. The latter are assumed to be edge dislocations distributed on three orthogonal slip planes with three orthogonal Burgers’ vectors in a simple cubic crystal. The dislocation densities are proportional to the sink strength [67] for each direction, are $\rho_{a_{1}}$, $\rho_{a_{2}}$ and $\rho_{a_{3}}$ and correspond with the three orthogonal axes a_1_, a_2_ and a_3_. The cavity sink strength is ρ_c_ and is given by 4 πrN, where r is the cavity radius and N is the number density [67]. Using the simple balance equations shown in the inset of Figure 30 the creep and swelling rates can be calculated.

Two cases will be considered: (i) steady-state dislocation density = 10^15^ m^−2^; (ii) steady-state dislocation density = 10^14^ m^−2^. These densities correspond with the steady-state values for fuel pins irradiated in DFR at temperatures of approximately 400 and 600 °C for 20% cold worked 316 SS [87,88]. It is assumed that the microstructure is isotropic and the dislocations are equally partitioned over the three directions. The creep and swelling rates are then calculated with and without an applied stress (assume equivalent to 100 MPa) but as a function of an evolving cavity structure consistent with the swelling shown in Figure 28. The calculated swelling rates, with and without an applied stress, and as a function of swelling, are shown in Figure 31. The results show that: (i) stress enhances the swelling; (ii) the swelling rate varies non-linearly as the swelling increases. The swelling rate is a function of the evolving cavity density and the dislocation density, as one might expect.

Figure 32 shows the creep strain partitioned in each of the three directions and compared with the full swelling strain and the linear swelling strain (calculated with the applied stress). As one might expect the creep strain is higher in the direction of the applied uniaxial stress and starts out negative in the other two directions until the swelling advances sufficiently to make all strains positive.

Figure 33 shows the derived value of the creep-swelling compliance, D (assuming that s = 0.15 corresponds with 100 MPa) as a function of swelling for the two steady-state dislocation densities. Also plotted are the corresponding swelling rates calculated for the same conditions used to derive the D values. The results are qualitatively the same as the experimentally measured D values (inset) showing that the simple balance equation construct illustrated in Figure 30 is a reasonable facsimile of real behaviour [95].

It is likely that D represents the strain arising directly from anisotropic diffusional mass transport subject to an applied stress, but it could also include an element of creep that is based on dislocation slip of network dislocations, i.e., stress-induced climb and glide (SICG) that is enhanced by swelling. The fact that there is only a weak sensitivity to the network dislocation density for irradiation creep of austenitic stainless steels, even in the swelling regime [3,68], implies that dislocation slip is not the dominant strain-producing mechanism. Climb of dislocations, either pre-existing network or newly created dislocations loops is then the most likely source of strain, both for swelling and irradiation creep. In this context, the difference in the way that strain is manifested by the glide of network dislocations created by shear, or the formation of dislocation loops that may also subsequently glide, may be important (see Section 7.4).

### 7.3. Creep Suppression

The irradiation creep of stainless steels exhibits primary, secondary and tertiary stages. In the context of Equation (12), the primary creep is subsumed into the B_0_ term and accounts for the decreasing creep rates with increasing dose at doses <5 dpa (Figure 29). In one sense irradiation creep can be considered as a continuum because the microstructure is always evolving, especially when cavity formation is occurring over large dose ranges. In the early part of what might be considered secondary creep the dislocation structure is evolving. In the latter part of secondary creep, the response is continuously changing as the cavity structure evolves. The cavity evolution is controlled by bias-driven growth and by the accumulation of He. In the case where the effect of He is low or negligible one can focus solely on the bias-driven growth. At a low stress, and at temperatures where swelling is active, the contribution of bias-driven diffusional mass transport to the secondary stage can, and in many cases will, terminate. Of course, the assumption is that the creep strain is low enough that one has not progressed to tertiary creep. Let us consider mass transport alone, without consideration of the effect of the accumulation of increasing amounts of He in the cavities with increasing dose.

The phenomenon of creep suppression has been observed and referred to as creep disappearance [3]. This is demonstrated in the experimental data as it applies to B_0_ (Figure 29b) and to D (inset of Figure 33). Termination of secondary creep can occur for many reasons depending on whether one is talking about swelling-independent or swelling-dependent creep. The SICG component of dislocation creep (B_0_) must end because the dislocation structure is either exhausted (gliding dislocations leave the crystal) or glide is blocked by an evolving cavity microstructure. For a similar reason, the diffusional creep rate (D) will be expected to pass through a peak as the microstructure reaches an optimum state due to the evolving cavity structure, diminishing thereafter (see Figure 31, Figure 32 and Figure 33). There will be no steady-state creep if the microstructure continues evolving unless the grain boundary sink strength is large, i.e., the grain size is small [68,75].

When there are other sinks besides cavities and dislocations (network and prismatic loops), such as grain boundaries, the boundaries can act as biased sinks due to elastodiffusion. In that case the creep can continue indefinitely. A high density of grain boundary sinks will also quickly limit that portion of the creep that is caused by dislocation glide. Once a dislocation has run into a grain boundary its contribution to creep stops. Therefore, a small grain size will quickly suppress creep by dislocation glide but will promote continued creep (no suppression) by elastodiffusion [68,75,94].

It may be argued that, even in the absence of yielding, there is an evolving dislocation structure because of the nucleation and growth of dislocation loops. However, prismatic dislocation loops are fundamentally different from dislocations created by shear and do not contribute to creep strain by the same mechanism (see Section 7.4).

### 7.4. Climb versus Glide

Irradiation creep strain involving dislocations is either caused by glide of dislocations that have been enabled by enhanced climb as a result of irradiation, or simply by the climb itself. Climb enabling glide of edge network dislocations [68,75] results in no increase in dislocation density, i.e., sink strength. Climb of prismatic dislocations loops, however, does involve an increase in dislocation density if the climb is in the direction of loop growth. For the most part, an evolving dislocation structure from dislocation loop formation and growth reaches a constant state at low doses that corresponds to a saturation in mechanical properties such as yield strength (Figure 26). The density (sink strength) of dislocations reaches a constant value at low doses (5–10 dpa) at which point the irradiation creep rate is more-or-less constant (Figure 29), although any long-term evolution in the microstructure will affect the creep, whether by the swelling-independent creep compliance (B_0_) or the creep-swelling compliance (D). Whether or not B_0_ is dependent on dislocation climb-enhanced glide or simply climb, the flux of point defects to enable the climb will be dependent on the same mass transport partitioning that governs D as we have seen in the previous section (Figure 30).

Network dislocations (those created by glide rather than the climb of dislocation loops) have the potential to contribute to creep strain by glide because the strain from these dislocations created by shear is the product ρ × b × l, where ρ is the dislocation density, b is the dislocation Burgers’ vector and l is the distance travelled by the gliding dislocation over the time of the creep measurement. Prismatic dislocation loops, however, have a different effect on creep strain even if they have un-faulted and grown so large that they are indistinguishable from a normal cold-worked dislocation network created from shear loops [3].

Unlike dislocations created by shear, the strain from prismatic dislocation loops is manifested as soon as the loop is formed and increases as the loops grow, even if they grow so large that they are indistinguishable from network dislocations created by shear. Irrespective of how large they may have grown, prismatic dislocations (loops) do not add to the strain simply by translation within their glide cylinders. The only influence that gliding dislocation loops can have on the strain exhibited by an irradiated material is by: (i) redistribution of the strain of the dislocation loop within the volume of the glide cylinder; and (ii) removal of the loop from the grain interior, thus creating more space for new loop evolution from point defect clustering and climb.

It is a common misconception that gliding dislocation loops, or the larger network created by coalescence of prismatic loops, can create additional creep strain through slip on their glide cylinders that is the same as the strain from shear (network) dislocations. Once a loop has formed, the strain is manifested in the material but may be distributed over the surface of the crystal or grain and any translation of the loop toward the edge of the grain or surface simply concentrates the strain over a smaller area of the surface nearest to the loop. Hypothetically, if a prismatic loop was to shear so that opposite sides move in opposite directions, there would be a redistribution of the prismatic loop strain over the volume of the glide cylinder but no change in the tensile strain parallel with the loop Burgers vector. Figure 34 illustrates the difference between the effect of a shear stress (σ_13_) acting on a dislocation source to create a shear loop (where the loop lies in the slip plane) compared with a shear stress acting on a prismatic loop (where the loop lies on a plane inclined to the glide cylinder defined by the Burgers vector). In Figure 34a, a dislocation source is activated at the centre of the crystal. Initially there is no strain but as the dislocations progress towards the boundary of the crystal the cancellation of the tensile and compressive strains above and below the slip planes of the dislocation segments of opposite sign diminishes. As the extra half planes glide towards the surface of the crystal, the elastic strain from the dislocation at the surface increases until the extra half plane reaches the surface and produces a step. In Figure 34b, the elastic strain on the boundary of the crystal from the prismatic loop is assumed to be constrained within the glide cylinder, although in reality it will be distributed over the whole crystal boundary and the strain at the surface of the crystal will only concentrate at the intersection with the glide cylinder as the prismatic loop gets close to the surface. If the loop shears and also translates within the volume of the glide cylinder the strain from the sheared segments will be partitioned between the crystal surfaces intersecting the glide cylinder, as shown in Figure 34b. If the tensile strain from the loop, due to the displacement defined by the Burgers’ vector (b), is defined as ε_11_, the additional shear within the volume of the glide cylinder can be defined as ε_13_. In both cases, the elastic strain at the crystal surface is shown by the light blue curves. For ease of illustration, the strain is assumed to be manifested perpendicular to the extra half-plane(s) and within the glide cylinder in both cases.

## 8. Creep Rupture

The irradiation creep of stainless steels exhibits primary, secondary and tertiary stages. It is the tertiary stage (the necking stage) that results in creep rupture and is a concern at high temperatures when thermal creep starts to dominate [96].

Non-swelling creep (B_0_), which one can attribute to SICG, is either weakly or negatively correlated with temperature, the latter because of the effect of temperature on recovery of the network dislocations as the temperature is increased [68]. The swelling creep compliance (D) is negatively correlated with swelling rate [12], as demonstrated in Figure 33. Due to the inter-relation with swelling, which is temperature dependent, it is difficult to know what the intrinsic effect of temperature is on D. At a sufficiently high temperature, the creep can be considered thermal, i.e., driven by dislocation slip, the same as it would be for an unirradiated material. However, for irradiated materials, the microstructure has changed, and the tertiary stage of creep cannot be expected to be insensitive to the effects of irradiation on the microstructure, in particular any loss of ductility caused by cavity segregation at grain boundaries.

Most creep rupture tests are conducted out-reactor after irradiation so we are really considering thermal creep tests on pre-irradiated material that will likely give the same result in-reactor, especially at temperatures >600 °C [96].

There are not yet sufficient data to determine the effect, if any, of helium directly on the irradiation creep itself. However, a reduction in ductility due to He embrittlement will affect the time to failure for a given tertiary creep rate. Failure will be expected at lower strains or lower true stresses in the He-embrittled material because of perforation of the grain boundaries (see Figure 16 and Figure 17). Failure in a creep rupture test corresponds to a lower engineering stress and strain for some irradiated materials when voids (cavities) are present and the fracture is typically inter-granular [97]. Creep rupture is exacerbated by helium since bubble formation at grain boundaries is the primary mechanism of in-reactor failure, especially at high temperatures [46].

Typically, one observes a reduction in both rupture times and stresses in out-reactor creep rupture tests indicating that the irradiation has affected the material in a way that is permanent. Materials that exhibit poor creep rupture properties tend to have high helium concentrations and exhibit inter-granular failure [43,46,97]. For a given stress state, the time to rupture is reduced by the accumulated dose, getting increasingly worse with doses up to 50 dpa. As the dislocation structure is more-or-less constant after approximately 5–10 dpa, even taking into account recovery [46,87,88], the strain before creep rupture happens appears to be sensitive to the accumulation of He-stabilised cavities on grain boundaries. The main feature that changes with irradiation is the cavity structure and the main thing that affects creep rupture is helium.

According to Grossbeck et al. [46], irradiation creep is a non-damaging process in the absence of helium. When helium is present it promotes the growth of He-bubbles or He-stabilised cavities on grain boundaries. The grain boundary coverage is exacerbated by a tensile stress acting on the boundary [43]. As the He accumulation is occurring during in situ creep testing, it is likely that the enhancement on boundaries perpendicular to the tensile stress is the result of preferential diffusion of interstitial He in directions parallel with the tensile stress but may also be because of enhanced growth of existing cavities subjected to the stress at high temperatures [38]. The degree to which the grain boundary coverage by cavities correlates with low ductility and strength in 316 SS has been demonstrated by Schroeder and Batfalsky [43], as shown in Figure 11. Samples of 316 SS were creep tested to failure at 750 °C during He implantation (“in-beam”), with implantation rates of 10–100 appm He/h. The times to rupture (t_R_) are given in the figure captions for the two cases cited in Figure 11. Although these low stress creep tests do not give the same information as a simple tensile test, from Figure 11 it is apparent that the time to rupture at a given stress is lower for the 316 SS alloy with the higher He concentration of 2500 appm, compared with 500 appm, at 750 °C.

## 9. Conclusions

Austenitic stainless steels have good ductility and strength, during and after irradiation in a power reactor, provided the temperature is not high (<350 °C) and the main form of irradiation damage is dislocations loops. At low doses, austenitic stainless steels exhibit an increase in yield strength and a reduction in ductility that is the result of clustering of point defects, dislocation loops in particular. The radiation damage structure lends itself to localised interactions with gliding dislocations that can sweep up the loops. The removal of dislocation loops creates a softer region promoting further easy deformation in channels. The localisation of the deformation in softer channels concentrates the strain in small volumes and reduces the ductility as a result. The fracture occurs at low strains but is still ductile. This channelling-softening effect is also observed in Zr alloys.

Unlike Zr alloys, austenitic stainless can suffer from He embrittlement at high doses and temperatures provided they operate in a neutron spectrum that produces sufficient helium from (*n*, α) reactions. At low thermal neutron doses (<1 × 10^21^ n cm^−2^, E < 0.5 eV), the main source of He is from the transmutation of ^10^B, which is present as an impurity in all austenitic stainless steels. There is a limit to the total amount of He produced based on [B], the helium concentration in appm being approximately equal to the boron concentration in wtppm. The He production is enhanced by Ni, which is approximately 10 wt% in 304 and 316 stainless steel, whether by high-energy (*n*, α) reactions or by reactions involving thermal neutrons (^58^Ni→^59^Ni→^56^Fe + He). The He produced in these cases can have a severe detrimental effect on the ductility of the material promoting inter-granular failure and poor creep rupture properties.

The dimensional stability of austenitic stainless steels is largely dependent on the swelling, which for the most part is also dependent on having He present to stabilise cavities against collapse to vacancy dislocation loops. The swelling that is observed increases with increasing temperature at reactor operating temperatures >250 °C. There is a peak in the swelling rate per unit dose at approximately 500 °C, decreasing thereafter at higher temperatures. Modelling shows that the temperature dependence is the result of the effect of temperature on the internal pressure within cavities (for a given He content), the vacancy point defect mobility and the ease of vacancy emission from the cavity free surface (dependent on surface energy and vacancy formation energy).

Irradiation creep is a complex process that involves the elevated concentration of freely-migrating point defects produced by irradiation that promotes the climb of network dislocations and dislocation loops. Climb itself contributes to the strain in a given direction under the action of a stress. This can either be considered simply in terms of work done or in terms of the diffusional drift of point defects so that climb of dislocations of a given Burgers vector is enhanced by elastodiffusion or SIPA. For enhanced climb and glide under the action of a stress (SICG), the hardening effect of the irradiation on the microstructure is important, effectively reducing the creep per unit dose in the early stages of irradiation until the loop density reaches a constant state and the creep from dislocations (from either climb or glide) reaches a steady-state condition. It is difficult to know how much of the creep strain comes directly from climb and how much by glide. In this context, the glide component must eventually be exhausted, either because the dislocations have travelled far enough to leave the crystal or if they are blocked by additional barriers to slip, i.e., cavities, that evolve over a longer period compared with the dislocation loop structure.

The irradiation creep processes that affect the swelling-independent creep compliance (B_0_) in the steady state can either be some form of enhanced glide (SICG) or enhanced climb (SIPA) or simply the effect of stress on the direction of diffusion (elastodiffusion). B_0_ is initially high and then decreases to a minimum steady-state condition after a dose of approximately 10 dpa. Up until the steady state is achieved (from 0 to 10 dpa) the main creep process is likely to be SICG. The accumulation of irradiation damage during this early stage of irradiation effectively blocks and inhibits the glide of network dislocations, thus reducing B_0_ as the radiation damage density increases. There is then a degree of ambiguity surrounding how much of the steady-state B_0_ is dependent on simple mass transport (because of elastodiffusion and SIPA) and how much by dislocation glide (SICG). Because the FMD production is largely athermal any temperature effects of irradiation creep appear to be manifested in the effect of temperature on the microstructure (dislocation density and even the cavity density). As the temperature increases, and the damage density diminishes, the creep from dislocation glide may be enhanced because the distance between the blocking centres increases. The inter-barrier distance effect is offset by the size-effect of the barrier itself, increasing the strength of the barrier (α).

The other component of irradiation creep (that which controls D) is very much dependent on diffusional mass transport because of the inter-relation with swelling. Swelling is dictated by the sink strength (size and number density) of the neutral sinks (cavities) that play an important role in controlling the partitioning of the point defects between the dislocations and the cavities. In the sink-dominated regime, because the FMD concentrations (vacancy and interstitial) are largely athermal, any temperature dependence of creep is likely to be because of the effect of temperature on the cavity stability, which is also dependent on the presence of He. For a given neutron flux, that part of irradiation creep contributing to D will diminish at temperatures >500 °C because of the decrease in swelling. In this temperature range creep due to dislocation slip (conventional thermal creep) will increase, provided there are sufficient gliding dislocations. Because diffusional mass transport depends on a balance of fluxes of interstitials and vacancies to different sinks, it is inevitable that the swelling creep compliance (D) is not independent of swelling

At high temperatures (550–650 °C), the reduction in the time, or the stress, for creep rupture is sensitive to He content, not because of the irradiation creep rate, which will be decreasing as the swelling rate decreases at higher temperatures, but because of the accumulation of He-stabilised cavities on grain boundaries. The accumulation of He on boundaries increases with increasing He content but also increases with temperature. The same processes resulting in He embrittlement apply to creep rupture. The presence of perforated grain boundaries due to He-stabilised cavities reduces the applied stress for a given time, or the time for a given applied stress, prior to rupture of pre-irradiated material or material tested during irradiation. The reduced ductility corresponds with inter-granular fracture because the grain boundaries are perforated by He-stabilised cavities that are sites for preferential failure.

## Figures and Tables

**Figure 1 materials-14-02622-f001:**
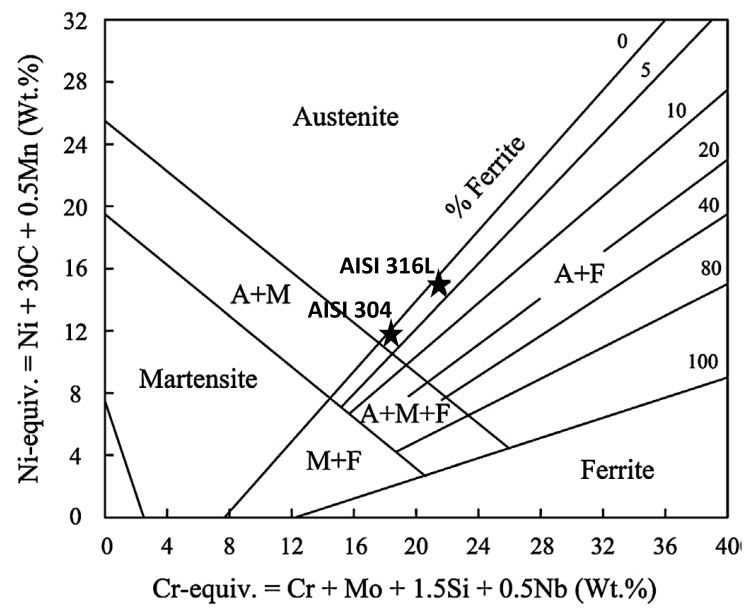
The Schaeffler diagram for Fe–Cr–Ni stainless steel structure predictions in welds. Modified from [2].

**Figure 2 materials-14-02622-f002:**
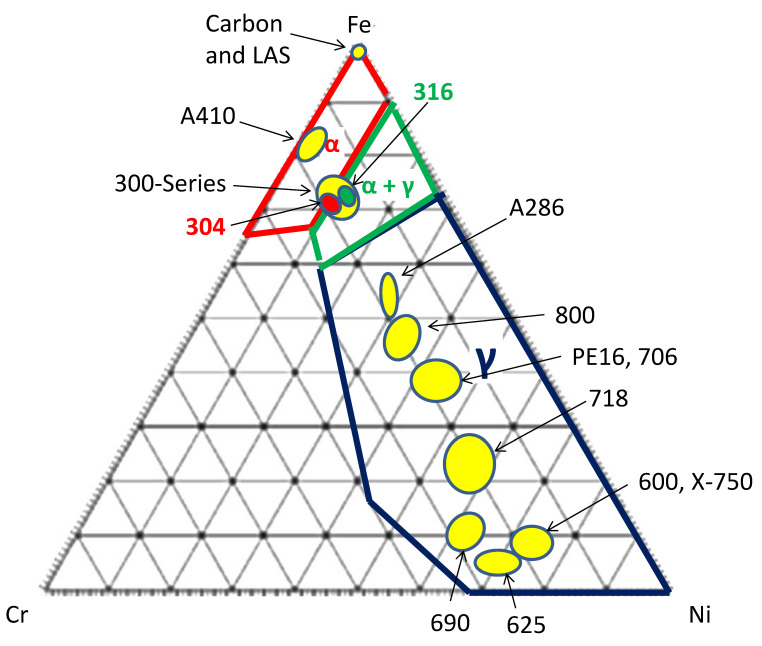
Ternary diagram for the Fe–Cr–Ni system at 400 °C illustrating the approximate compositions (considering Fe + Cr + Ni = 100 at%) of the main structural alloys used in nuclear reactors. The composition range for the stable FCC phase (γ) is outlined in blue. The green region can be metastable, leading to a wide range of possible austenitic, ferritic and martensitic phases that make up the duplex stainless steels. The ferritic BCC phase (α) in the red region illustrates the composition range for the high-strength carbon and low-alloy steels (LAS) and ferritic stainless steels used for pressure vessels and piping.

**Figure 3 materials-14-02622-f003:**
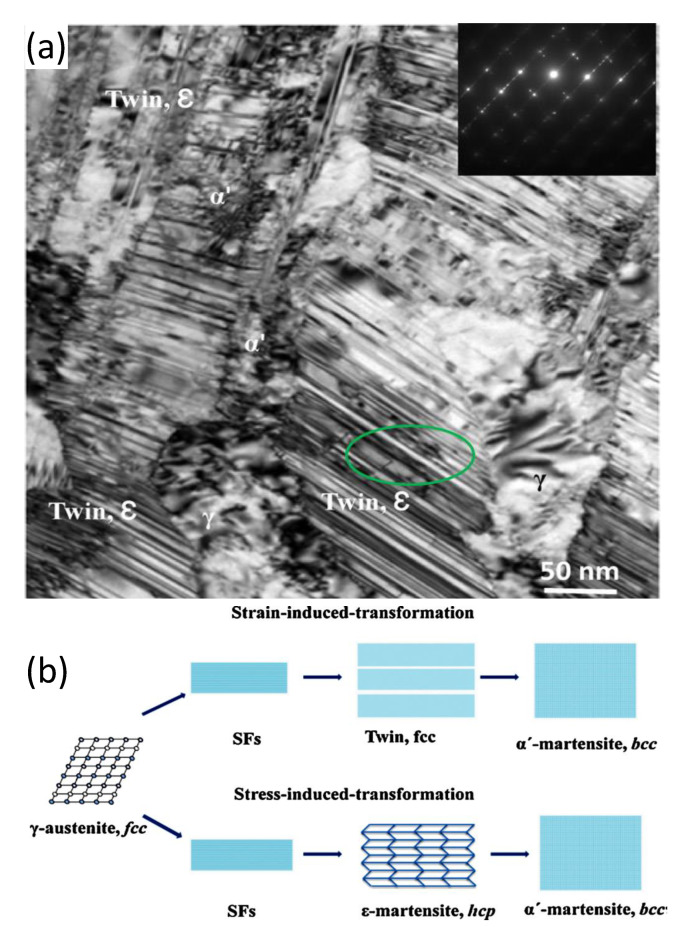
(**a**) TEM micrograph and (**b**) schematic diagram, depicting how the FCC γ phase transforms to the BCC α’ phase via planar shear creating stacking faults followed by bulk shear of twins and/or ε-martensite as intermediary stages. Modified from [6].

**Figure 4 materials-14-02622-f004:**
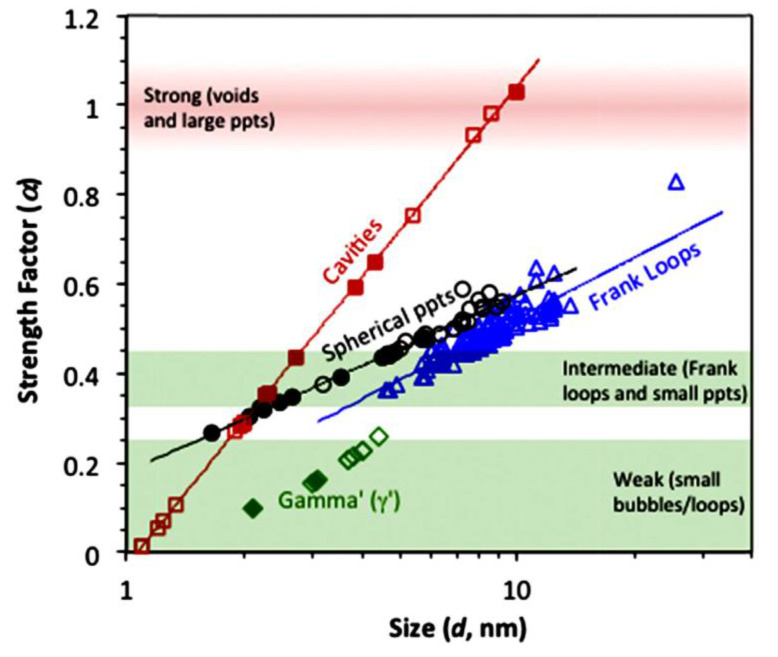
Calculated size-dependent strength factor at room temperature for spherical incoherent precipitates ppts (circles), coherent ppts (diamonds), Frank loops (triangles), and cavities (squares). Reproduced with permission from [15].

**Figure 5 materials-14-02622-f005:**
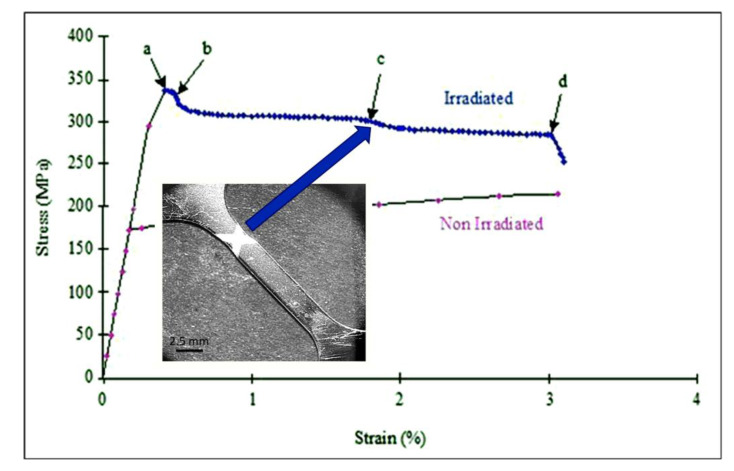
Engineering tensile stress–strain curve for a Zircaloy-2 sheet that had been irradiated at 300 °C to a neutron fluence of 1.1 × 1025 nm^−2^ and subsequently tested at 25 °C at a strain rate of 4.7 × 10^−6^ s^−1^. For description of points (a–d), see text. Modified from [17].

**Figure 6 materials-14-02622-f006:**
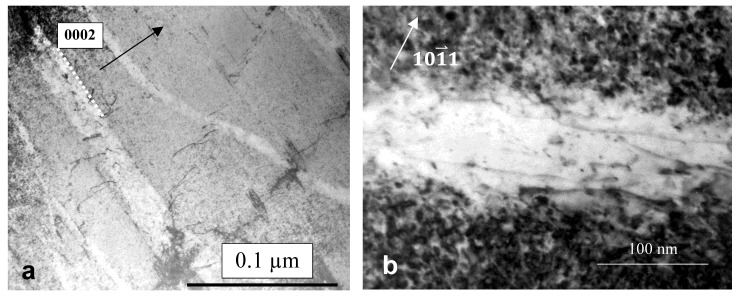
Dislocation channels in a Zircaloy-2 plate that had been irradiated at approximately 300 °C and tested in uniaxial tension at room temperature: (**a**) micrograph showing channels parallel and inclined to the basal plane; (**b**) high-magnification image showing dislocation loops and network dislocations within a channel.

**Figure 7 materials-14-02622-f007:**
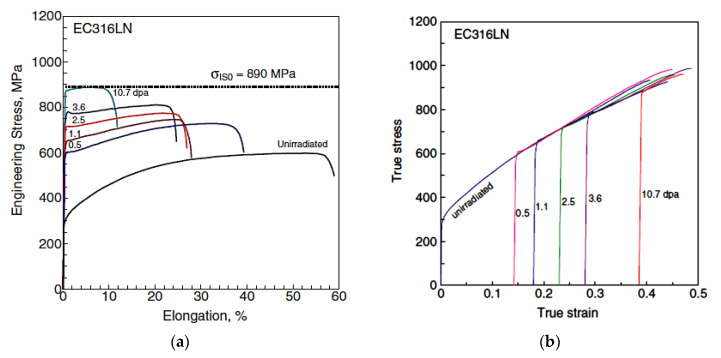
(**a**) Engineering stress–strain curves and (**b**) true stress—true strain curves for EC316LN stainless steel irradiated at 60–150 °C and tested at room temperature. The true stress—true strain curves of irradiated specimens are shifted in the positive direction by strains of 0.14, 0.18, 0.23, 0.28, and 0.385, respectively, to superimpose on the curve of unirradiated material. Irradiation-induced increases in yield stress were 305, 358, 421, 485, and 587 MPa, respectively. Note load drops at the yield point for the irradiated specimens. Modified from [25].

**Figure 8 materials-14-02622-f008:**
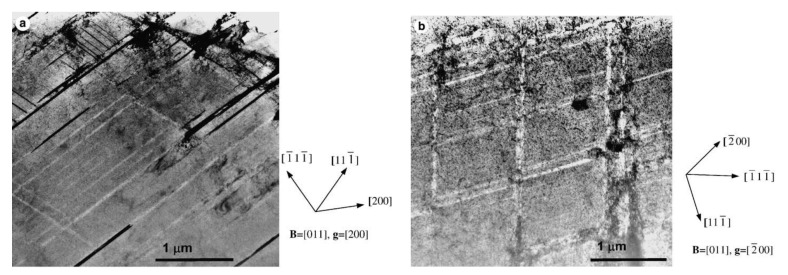
Examples of strain localisation and deformation twins in neutron-irradiated 316 SS after tensile testing: (**a**) mixture of twins and dislocation channels in 316 stainless steel irradiated.to 0.15 dpa and strained 6%; (**b**) intersecting arrangement of dislocation channels in 316 stainless steel irradiated to 0.78 dpa and strained 32%. Modified from [19].

**Figure 9 materials-14-02622-f009:**
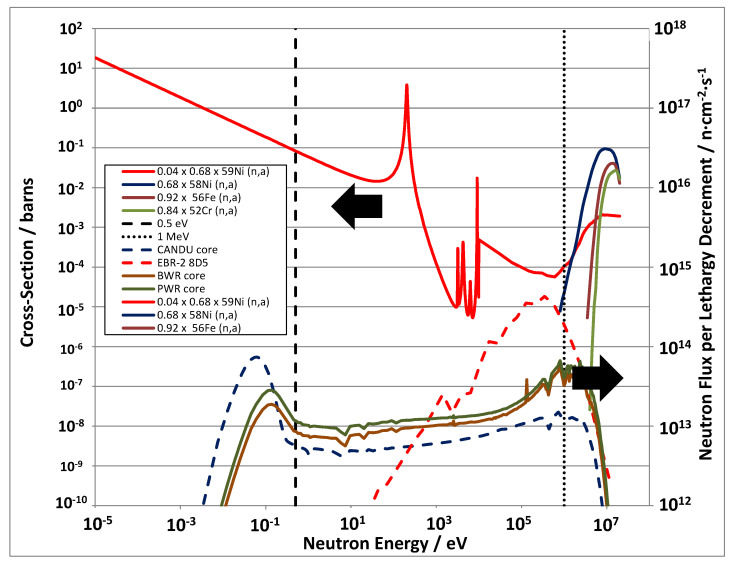
Neutron spectra for PWR, BWR, CANDU and EBR-II reactors (right hand scale) and (*n*, α) cross-sections for the major isotopes of Ni, Fe and Cr scaled for atomic abundance (left hand scale). Also shown is the ^59^Ni (*n*, α) reaction cross-section scaled for atomic abundance after 5 years operation in a CANDU core.

**Figure 10 materials-14-02622-f010:**
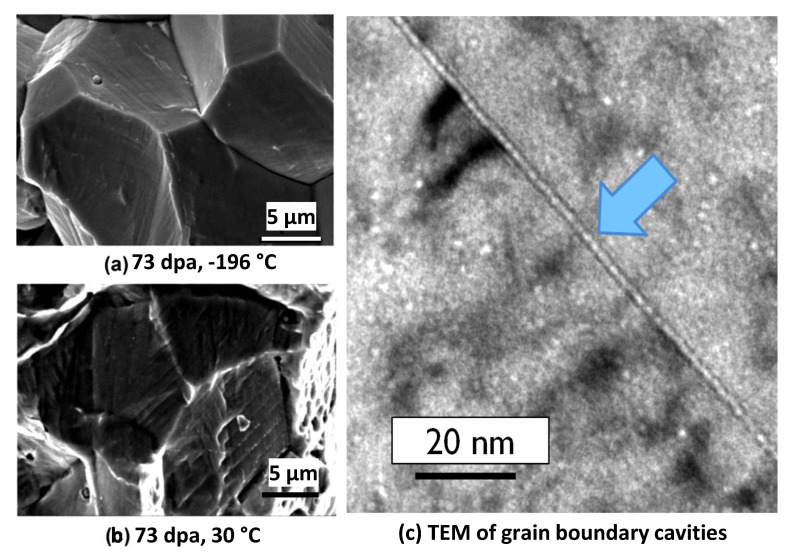
316 stainless steel irradiated in a PWR to approximately 70 dpa at 315 °C and containing 625 appm He:(**a**) fracture surface after impact test at −196 °C; (**b**) fracture surface after impact test at −30 °C; modified from [39]; (**c**) TEM micrograph showing grain boundary cavity segregation (arrowed); modified from [40].

**Figure 11 materials-14-02622-f011:**
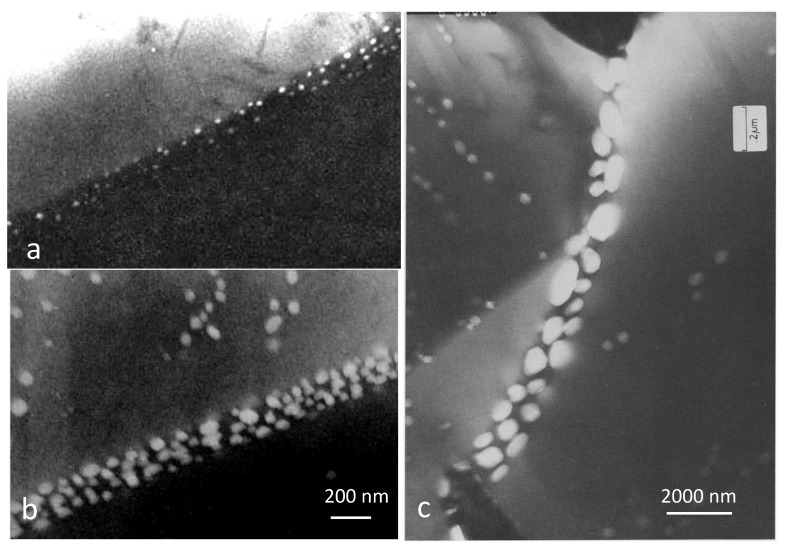
Comparison of bubble microstructures (TEM) for two samples of AISI 316 SS “in-beam” tested in tension to rupture at 1023 K at different stress levels. (**a**) σ = 77 MPa; t_R_ = 5 h; [He] = 500 appm He; mean bubble diameter: 20 nm. (**b**) σ = 50 MPa: t_R_ = 25 h; [He] = 2500 appm He: mean bubble diameter: 50 nm. (**c**) Same conditions as (**b**) showing preferential accumulation on a boundary perpendicular to a horizontal tensile stress. Modified from [43].

**Figure 12 materials-14-02622-f012:**
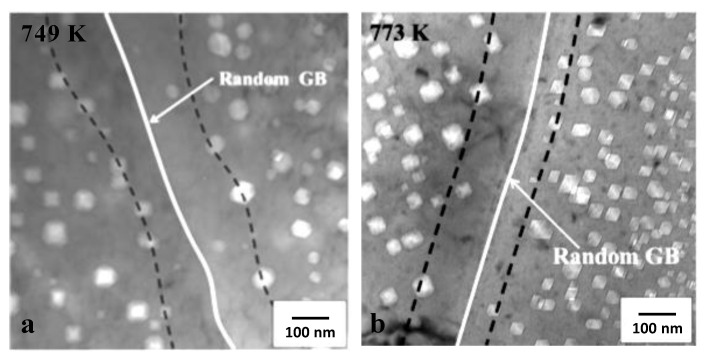
(**a**) Void distributions near the random GB in the Fe–15Cr–15Ni steel neutronn-irradiated in a fast reactor at 749 K to 18 dpa; (**b**) void distributions near the GB in the Fe–15Cr–15Ni steel electron-irradiated at 773 K up to 10.8 dpa. Modified from [44].

**Figure 13 materials-14-02622-f013:**
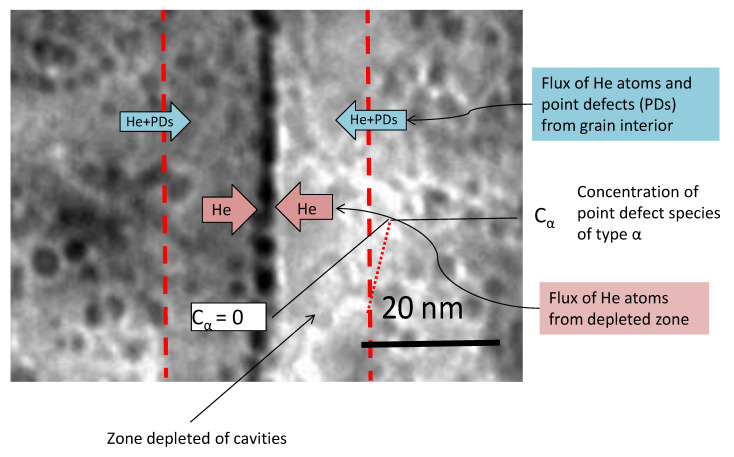
Micrograph illustrating cavity structure in neutron-irradiated Inconel X-750 and the source of He flow to grain boundaries taking into account any denuded zone. The He diffusing to the grain boundary comes from two sources: (i) the denuded zone where there is no trapping by matrix cavities; (ii) the grain interior where the flux is dependent on the internal sink structure. The red dotted line denotes the concentration profile up to the boundary that would normally exist for an irradiated material without a denuded zone, denoted by the red dashed line.

**Figure 14 materials-14-02622-f014:**
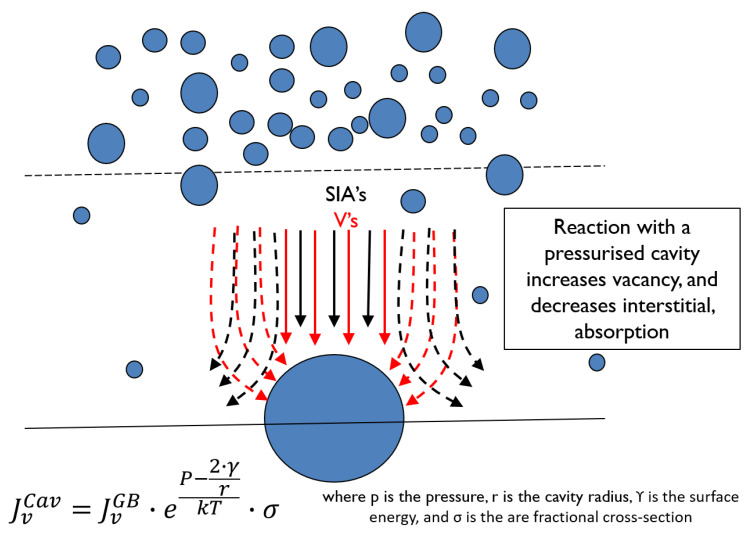
Schematic diagram illustrating source of vacancy (red) and interstitial (black) point defect fluxes to grain boundaries. There is a higher vacancy, relative to interstitial, flux from the grain interior. The flux to cavities is a function of the point defect flux to the boundary and an interaction term. p is the pressure, *r* is the cavity radius, γ is the surface energy, and *σ* is the area fractional cross-section.

**Figure 15 materials-14-02622-f015:**
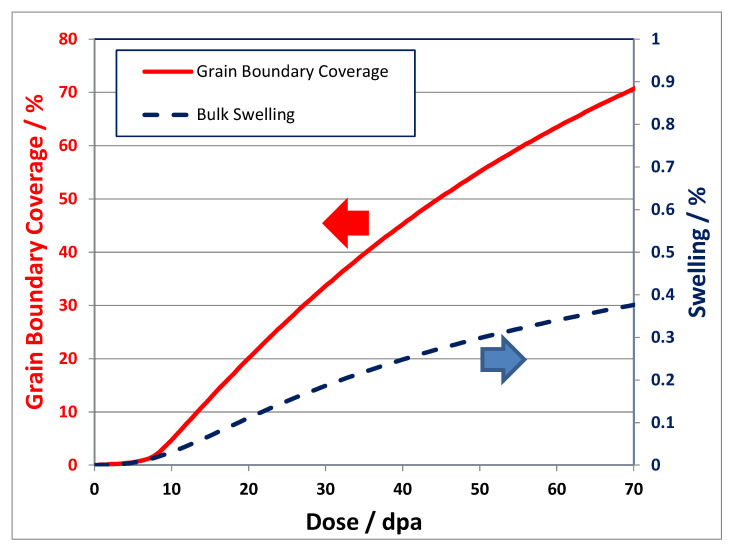
Calculated swelling and grain boundary area coverage for 316 SS as a function of dose (dpa) for a PWR flux thimble studied by Fukuya et al. [39] and Edwards et al. [42].

**Figure 16 materials-14-02622-f016:**
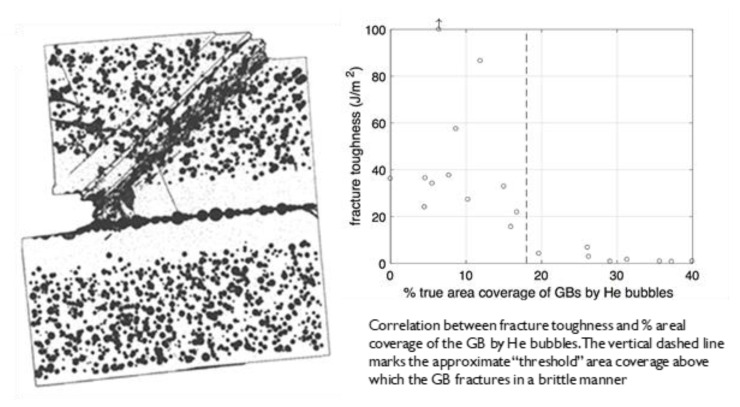
Calculated fracture toughness of irradiated Inconel X-750 as a function of % area coverage by cavities. Courtesy, Demkowicz, M.J.

**Figure 17 materials-14-02622-f017:**
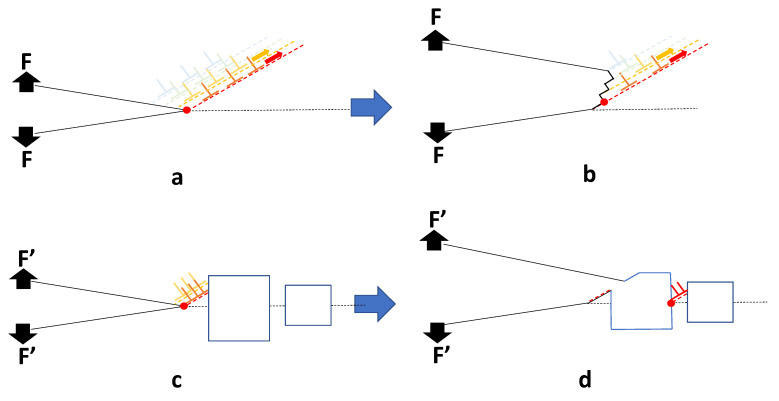
Schematic diagram showing the effect of cavities in restricting the volume for dislocation translation, thus limiting the absorbed energy for crack advance (lowering fracture toughness). The stress at the hinge points (red dots) will be dependent on the area of the ligament up to the nearest free surface (cavity): (**a**) crack on a boundary/interface subject to a crack-opening load (F); (**b**) crack blunting and energy absorption due to dislocation emission; (**c**) restricted dislocation emission (lower energy absorption for crack advance) and lower applied force (F’) needed to activate slip in the presence of cavities; (**d**) hinge point shift to next ligament.

**Figure 18 materials-14-02622-f018:**
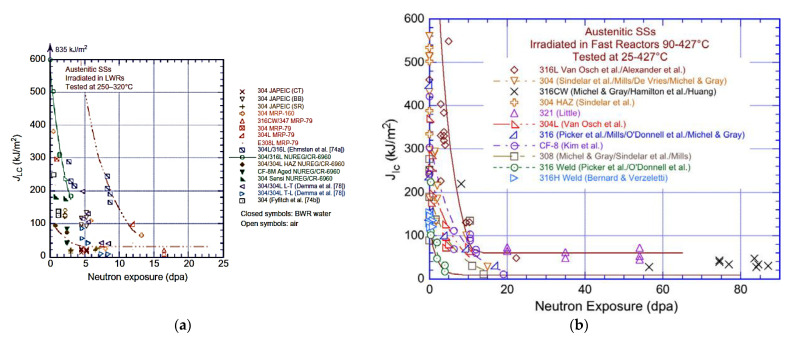
Fracture toughness as a function of neutron dose for austenitic alloys irradiated in: (**a**) fast reactors, 90–427 °C and tested in the temperature range 25–427 °C; (**b**) LWRs, irradiated at 288–316 °C and tested in the temperature range 250–320 °C. Reproduced with permission from [54].

**Figure 19 materials-14-02622-f019:**
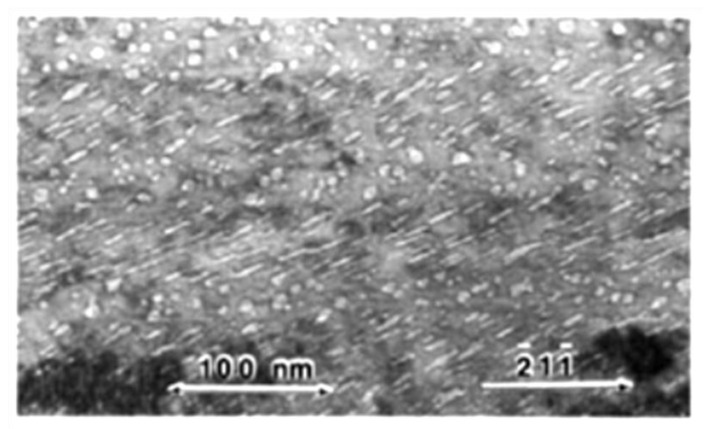
Sheared cavities in deformation twins in ion irradiated stainless steel. The twinning direction is $\bar{2}1\bar{1}$ as indicated on the micrograph. Reproduced with permission from [59].

**Figure 20 materials-14-02622-f020:**
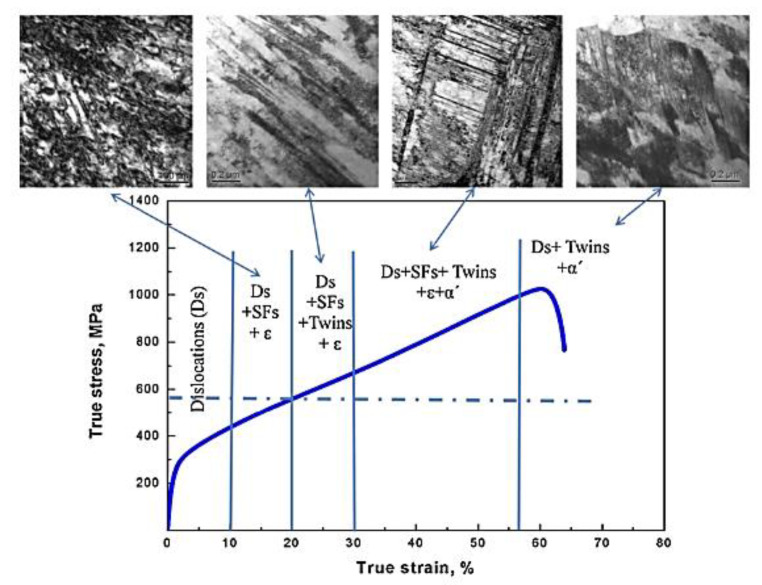
Progression of martensitic structures in 304 SS observed after room temperature deformation. The twin and ε-martensite phases tend to occur in platelets interspersed with one another. The text in the stress-strain plot denotes major forms of defects created during deformation: dislocations (Ds), stacking-faults (SFs), twins, ε-martensite and α’-martensite. Reproduced with permission from [6].

**Figure 21 materials-14-02622-f021:**
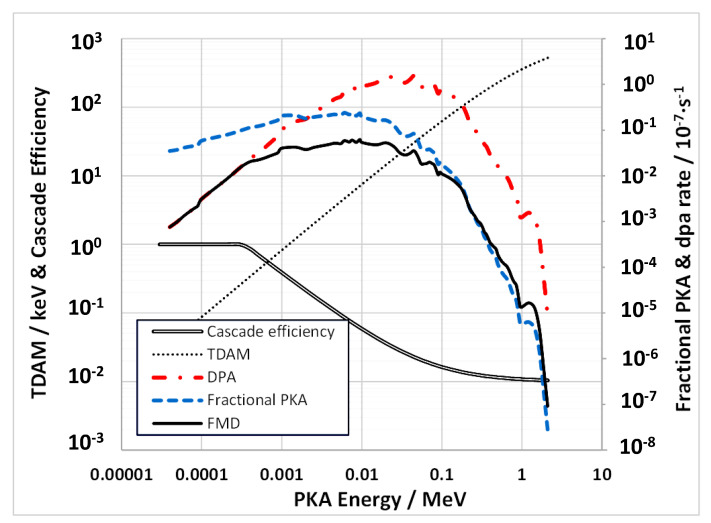
PKA spectrum, DPA and FMD for pure Fe in EBR-II 8D5 as a function of cascade energies for the different PKA values.

**Figure 22 materials-14-02622-f022:**
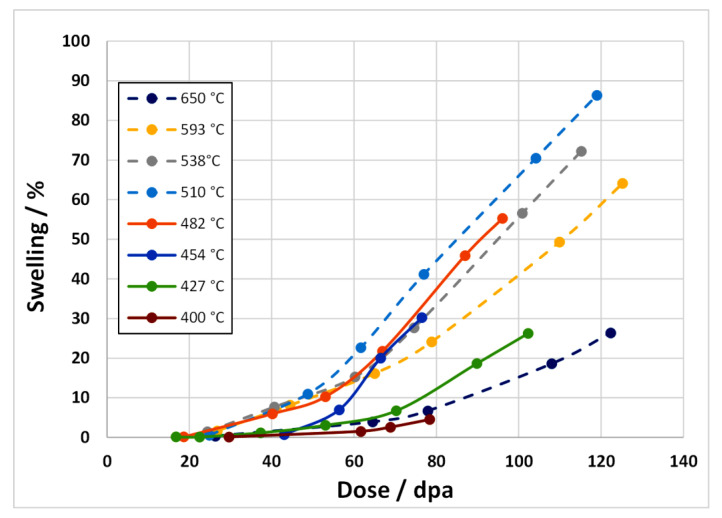
Swelling determined by density change as a function of irradiation temperature and dose, as observed in 20% cold-worked AISI 316 irradiated in the EBR-II fast reactor. All measurements at a given temperature were made on the same specimen after multiple exposures with subsequent reinsertion into the reactor. This procedure minimized specimen-to-specimen data scatter and assisted in a clear visualization of the post-transient swelling rate. Modified from [3].

**Figure 23 materials-14-02622-f023:**
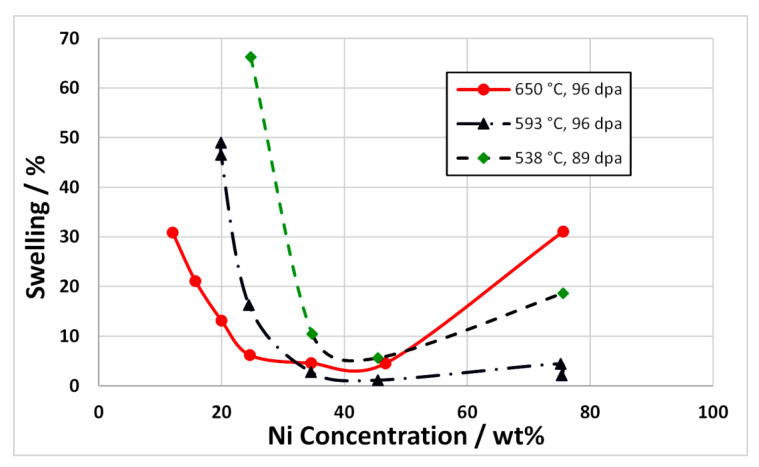
Dependence of swelling on nickel content of Fe–15Cr–Ni ternary alloys irradiated in EBR-II to doses of approximately 100 dpa, showing a minimum in swelling in the 30–50 Ni range, depending on temperature. Modified from [3].

**Figure 24 materials-14-02622-f024:**
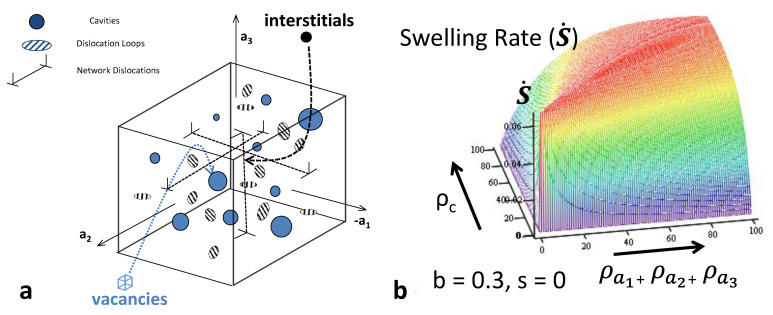
(**a**) Schematic of microstructure and (**b**) rate theory output showing irradiation swelling rate as a function of the relative dislocation (ρ_d_) and cavity/void (ρ_c_) sink densities/strengths in units of m^−2^. The volume swelling rate (% per unit dpa) is a maximum when $\sqrt{1+b}$ · ρ_d_ = ρ_c_, where *b* is governed by the size-effect interstitial bias for dislocations (**b**). The output shown is for a nominal 1% cascade damage efficiency and *b* = 0.3.

**Figure 25 materials-14-02622-f025:**
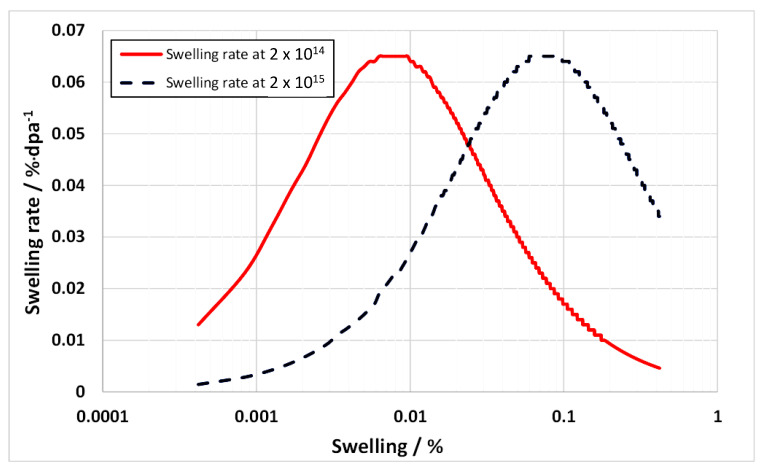
Calculated volume swelling rate (% per unit dpa) for nominal dislocation densities (ρ_d_) of 2 × 10^14^ m^−2^ (dotted curve) and 2 × 10^15^ m^−2^ (dashed curve), spanning the likely range of dislocation densities for an unirradiated and irradiated alloy, as a function of void number density and a mean void radius of 1 nm. The output shown is for a nominal 1% cascade damage efficiency and *b* = 0.3. Modified from [38].

**Figure 26 materials-14-02622-f026:**
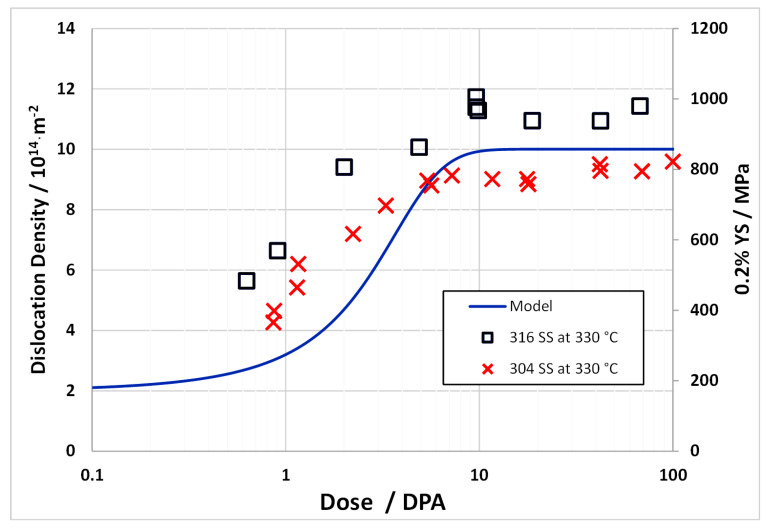
Uniaxial tensile yield stress data used for calculating swelling for austenitic stainless steels (316 SS and 304 SS) irradiated and tested at 330 °C. Modified from [68].

**Figure 27 materials-14-02622-f027:**
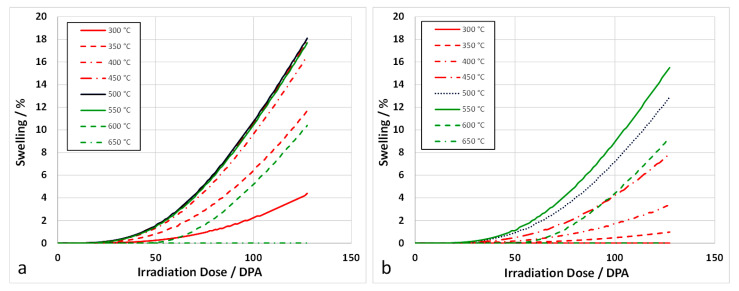
Calculated swelling for 316 SS as a function of dose in EBR-II (4.38 dpa per 10^22^ n cm^−2^) for a cavity nucleation density of 10^20^ m^−3^ and recombination coefficients of: (**a**) 10^20^ × D_i_ and (**b**) 100 × 10^20^ × D_i_, where D_i_ is the interstitial diffusion coefficient. The responses for different irradiation temperatures are illustrated. The dislocation density rises from ρ_d_ = 2 × 10^14^ m^−2^ in the cold-worked condition to 10 × 10^14^ m^−2^ after irradiation to approximately 10 dpa. FMD production rate = 2.52 × 10^−8^ dpa s^−1^, He production rate ~0.27 appm/dpa; 30% interstitial bias for dislocations, vacancy formation energy = 1.6 eV, vacancy migration energy = 1.4 eV, interstitial migration energy = 0.15 eV.

**Figure 28 materials-14-02622-f028:**
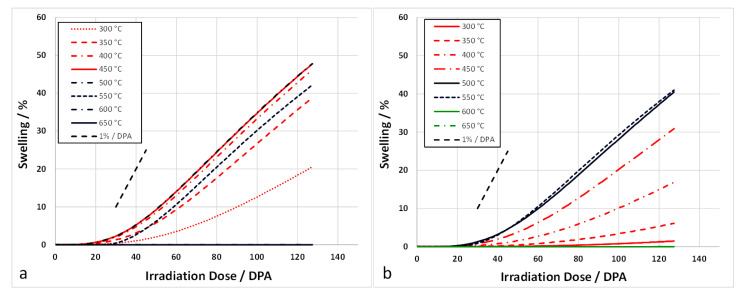
Calculated swelling of for 316 SS as a function of dose (DPA) at various temperatures in EBR-II for a cavity nucleation density of 10^21^ m^−3^ and recombination coefficients of: (**a**) 10^20^ × D_i_ and (**b**) 100 × 10^20^ × D_i_, where D_i_ is the interstitial diffusion coefficient. The dislocation density rises from ρ_d_ = 2 × 10^14^ m^−2^ in the cold-worked condition to 10 × 10^14^ m^−2^ after irradiation to approximately 10 dpa. FMD production rate = 2.52 × 10^−8^ dpa s^−1^, He production rate ~0.27 appm/dpa, 30%, interstitial bias for dislocations, vacancy formation energy = 1.6 eV, vacancy migration energy = 1.4 eV, and interstitial migration energy = 0.15 eV.

**Figure 29 materials-14-02622-f029:**
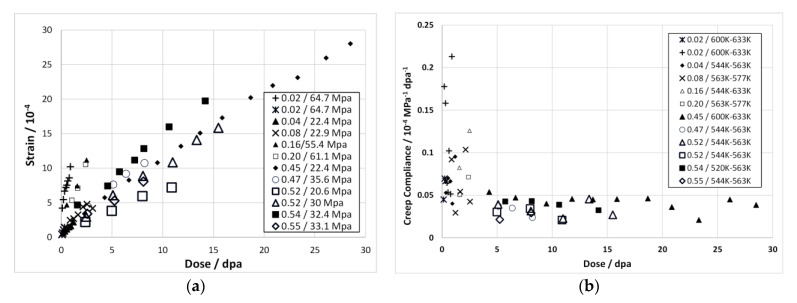
Creep strain and compliance as a function of dose rate for 316 SS springs irradiated in Dounreay fast reactors at temperatures between 520 and 633 K (247–360 °C): (**a**) strain as a function of dpa dose; (**b**) compliance as a function of dpa dose. The legend shows the dpa rate (in units of 10^−7^ dpa s^−1^) and stress for each dataset. Modified from [68].

**Figure 30 materials-14-02622-f030:**
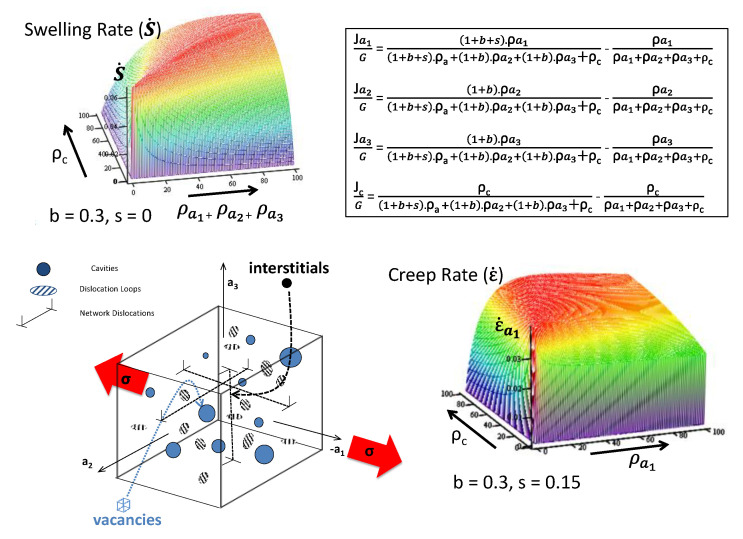
Simple rate theory formulation for calculating steady-state swelling and uniaxial creep rates as a function of the microstructure evolution. In this example, the dislocation density is chosen to be 10^15^ m^−2^, which is the expected density from residual cold work and radiation damage (dislocation loops).

**Figure 31 materials-14-02622-f031:**
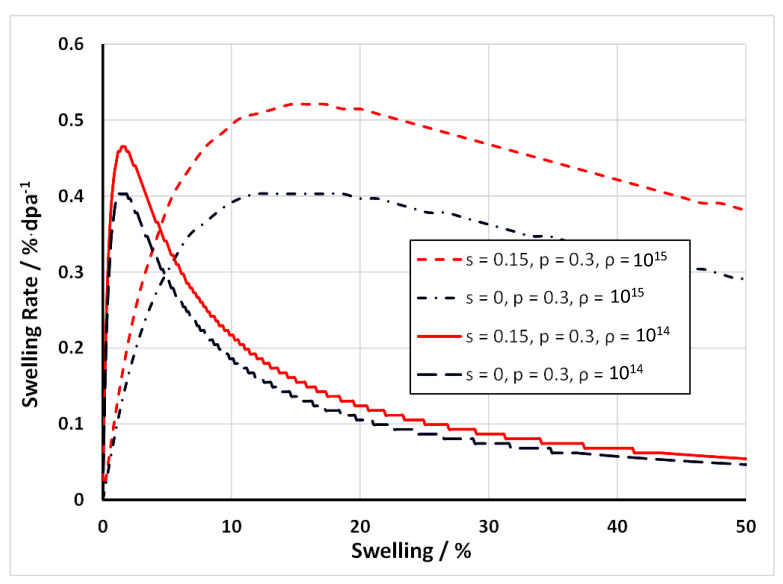
Swelling with (s = 0.15) and without (s = 0) stress for a dislocation density of 10^15^ m^−2^ and of 10^14^ m^−2^. It is apparent that swelling is enhanced by stress/creep.

**Figure 32 materials-14-02622-f032:**
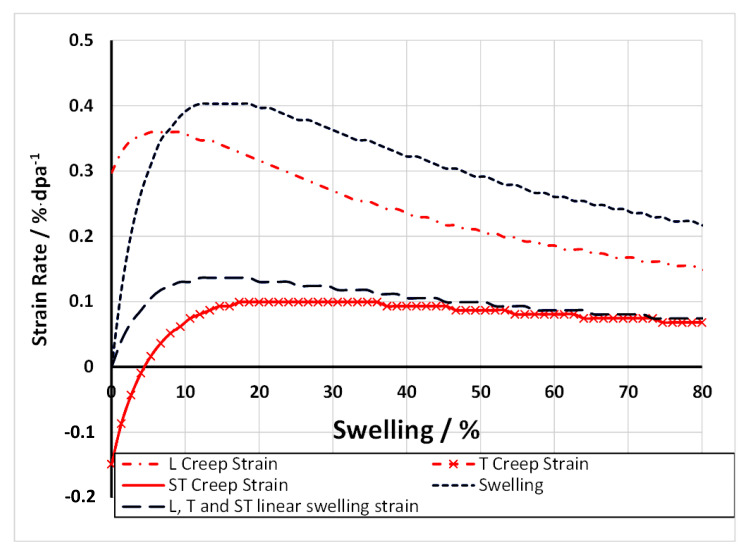
Uniaxial creep (L direction) and swelling for a dislocation density of 10^15^ m^−2^ s = 0.15, *p* = 0.3. L, T and ST refer to the longitudinal, transverse and short-transverse directions of a typical tensile specimen.

**Figure 33 materials-14-02622-f033:**
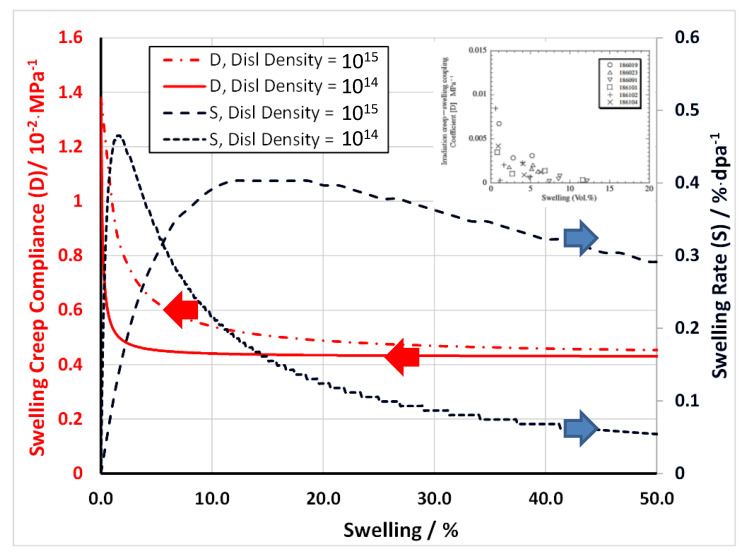
Creep Compliance and swelling under stress for a dislocation density of 10^15^ m^−2^ and of 10^14^ m^−2^. Assumes s = 0.15 is equivalent to 100 MPa. Inset—experimental data for D [95], corresponding with the calculated values for D shown in red.

**Figure 34 materials-14-02622-f034:**
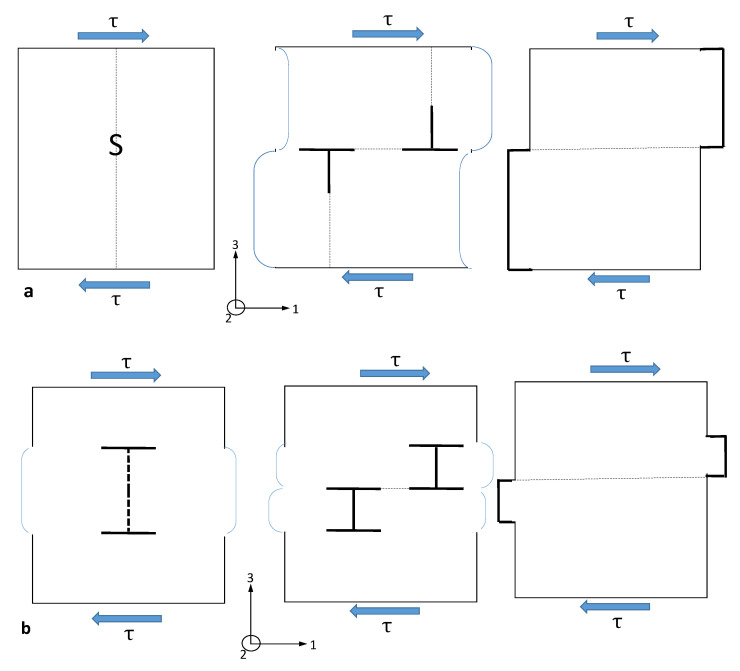
Schematic diagrams showing: (**a**) the translation of the extra half planes created by the activation of a shear source (S); and (**b**) hypothetical shear of a prismatic interstitial loop. Both views are projections perpendicular to the direction of the Burgers vector. The elastic strains at the surface of the crystal are indicated by the light blue brackets.

**Table 1 materials-14-02622-t001:** Main constituents of commercial austenitic stainless steels used in nuclear reactors (wt%). L refers to low carbon and LN refers to low nitrogen.

Alloy	Cr	Ni	Mo	Mn	Si	C	Ti	Nb	Others
286	13.5–16	24–27	1–1.5	<2	<1	0.03–0.08	1.9–2.3	4.7–5.5	-
304	18–20	8–12	-	1–2	0.75	0.08	-	-	304 L (0.03 C), 304 LN (<0.14 N)
316	16–18	10–14	2–3	2	0.75	0.08	-	-	316 L (0.03 C), 316 LN (<0.14 N)
321	17–19	9–13	-	1–2	0.75	0.04–0.08	0.2–0.4	-	(0.03 C, <0.14 N)
347	17–19	9–13	-	1–2	0.75	0.04–0.08		0.4–0.8	(0.03 C, <0.14 N)

**Table 2 materials-14-02622-t002:** Atomic displacement per atom (DPA) and freely-migrating point defect (FMD) production rates (atom fraction·s^−1^) for 316 and 304 Stainless Steel in EBR-II.

316 SS				304 SS			
Element	At %	DPA	FMD	Element	At %	DPA	FMD
Fe	70	2.72 × 10^−7^	1.61 × 10^−8^	Fe	71	2.77 × 10^−7^	1.64 × 10^−8^
Ni	12	5.42 × 10^−8^	4.27 × 10^−9^	Ni	10	4.54 × 10^−8^	3.57 × 10^−9^
Cr	17	7.27 × 10^−8^	4.27 × 10^−9^	Cr	19	8.16 × 10^−8^	4.79 × 10^−9^
Mo	1	4.71 × 10^−9^	5.73 × 10^−10^	-	-	-	-
Total	-	4.04 × 10^−7^	2.52 × 10^−8^	-	-	4.04 × 10^−7^	2.48 × 10^−8^

## Data Availability

Data sharing is not applicable to this article.

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
