# Peer review of "Effect of Neutron Irradiation on the Mechanical Properties, Swelling and Creep of Austenitic Stainless Steels"

_materials, 2021, doi:10.3390/ma14102622_

Round 1

Reviewer 1 Report

The manuscript needs to be sorted as a review paper not an article since majority of figures and data are from references. There is no new experimental or modeling results that purely come from the author.

References should be numbered following the Materials journal style.

Table and figure labels (e.g. 2-1) should also follow the Materials journal style in the text.

Author Response

Thank you for your time in reviewing this submission.

The original request from MDPI was to supply a submission to a special issue dealing with high temperature creep and mechanical properties of stainless steels.  As my expertise is in nuclear materials I proposed a submission on austenitic steels in nuclear reactors, which was acceptable to the journal editorial staff.  As the readers of the special issue would likely not be familiar with the pertinent literature I chose to both review relevant information on creep and mechanical properties and to submit new calculations of creep/swelling and microstructure evolution using conventional rate-theory.

The author is aware that the submission is part review and part new results (from modelling and calculations that apply to selected published data).  As the new results are from calculations applied to published experimental data, the format is more typical of a review.  Original data are shown in Figures 2-3, 4-4, 4-5, 4-6, 4-8, 6-1, 6-6, 6-7. 6-8, 7-2, 7-3, 7-4, 7-5, 7-6 (apologies the submission was missing a Figure 7-6 so in the revised version Figure 7-7 now becomes 7-6).

New calculated data for FMD and He-production in EBR-II are shown in Table 6-1.  There is also original work, previously published but modified in Figures 3-3, 6-4, 6-5, 7-1.

The manuscript has been revised to conform with the journal style.

Reviewer 2 Report

The manuscript contains a review of publications devoted to experimental research of the effect of neutron irradiation on the mechanical properties and creep of austenitic stainless steels. The manuscript also discusses the regularities of the formation of localized softening (dislocation channels) in zirconium alloys.

The manuscript also discusses the laws governing the formation of localized softening (dislocation channels) in zirconium alloys. Based on the analysis of the published data, the authors formulated the regularities of swelling of stainless steels under neutron irradiation, changes in their strength characteristics during creep at elevated temperatures and irradiation.

The quality of the manuscript should be significantly improved. The reviewer recommends making a correction to the text of the article to improve the correctness of the presentation of the results of data analysis. 

  1. (Page 10) The loading condition (tension or compression) have to be specified for stress-strain curve of the Zircaloy-2 alloy shown in Fig. 3.2. The test strain rate should also be specified.
  2. (Page 12) The loading condition (tension or compression) have to be specified for stress-strain curve of the EC316LN stainless steel shown in Fig. 3.4. The test strain rate should also be specified.
  3. (Pages 19 and 20) There are mistakes in figure 4.7 and 4.9. Dimension of the fracture toughness KIC is (MPa m1/2). See, please, ASTM E1820 Standard Test Method for Measurement of Fracture Toughness; ASTM E1823 Terminology Relating to Fatigue and Fracture Testing; ISO 12135 Metallic materials — Unified method of test for the determination of quasistatic fracture toughness. If figure 4.7 demonstrates of the material's notch toughness, the method of its determination should be indicated (Charpy impact test, Izod impact strength test etc.).
  4. (All Pages) Authors should adhere to a uniform style when designing illustrations. When specifying dimensional parameters on the axes of figures, different styles are used.
  5. (Page 29) It is necessary to indicate the correct dimension of the swelling rate in Figure 6.4 (b). Authors should also indicate the dimensions of relative dislocation (ρd) and cavity (ρc) sink densities/strengths.
  6. (Page 31) The loading condition (tension or compression) should be specified for the Yield stress values shown in Figure 6.6.
  7. (All pages) Authors should check the explanations of the parameters used in the formulas (2)-(17). Note, the symbols σd and εd are not present in the formulas of formulas (12) and (13).
  8. (Page 35). Authors should be made correction in the text to figure 7.1. The authors indicated that the figures demonstrate Creep as a function of dose rate, but in fact show (a) strain as a function of dpa dose; (b) compliance as a function of dpa dose.
  9. (Page 37).Authors should indicate the dimension of the parameters in figures 7.2.
  10. (Page 3.8). There is a mistake in figure 7.4. Strain rate dimension is not correct.
  11. (Pages 44-52). Authors should check the correctness of bibliographic cites in the reference list.

Author Response

Thank you for your insightful comments.  Apologies for adding data from Zr-alloys to help illustrate the strain localisation and softening processes that are observed in common with irradiated austensitic stainless steels.  The processes are the same but are better illustrated from work on Zr-alloys.

Regarding the specific comments:

  1. Loading orientation (tension) and strain rate have now been added in the figure caption.
  2. Loading orientation (tension) and strain rate have now been added in the figure caption.
  3. I agree, the conventional (ASTM) engineering measure of fracture toughness is a function of the crack geometry and the applied stress. In this case a physics-based measurement is used, i.e. the energy absorbed to advance the crack. The relationship between the two is described in Chopra and Rao (2011).  The engineering measure of fracture toughness (KIC) is concerned with the critical point where unstable crack growth occurs.  The value of KIC in the ASTM standard takes into account the geometry of the crack and the specimen for a given applied stress and has units of MPa.m1/2.  J, on the other hand, defines the resistance of the material to stable crack extension – how much energy is input to advance the crack.  Chopra and Rao (2011) show that the two are related,

The values of the fracture toughness in Figure 4-7 are the energy release rates as a function of crack area extension and are derived in “Computing critical energy release rates for fracture in atomistic simulations”, G.Q. Xua, M.J. Demkowicz, Computational Materials Science, Volume 181, August 2020, 109738.  A discussion of the modes of representing fracture toughness is beyond the scope of this paper but text has been added concerning the definitions of fracture toughness with appropriate references.  

  1. Noted – style will be normalised to conform with the journal guidelines.
  2. Swelling is dimensionless as it refers to Δm3/m3 - one option is to present this quantity as a % rather than a fraction, which is the unit adopted. Units have been added in the caption for Figure 6-4.
  3. Noted and added.
  4. Noted – the following text has been added – “The subscripts in these parameters refer to dislocations (d) and cavities (c).”
  5. Noted – the following text has been added – “Creep strain and compliance….”
  6. Both swelling and creep are dimensionless quantities. They are, by default fractional quantities – increase in volume per unit volume or linear dimension per unit dimension. They could sometimes be represented as a % but that would complicate the figures in this case.
  7. The dimension refers to the strain (in this case represented by %) per unit increase in displacement dose (represented by dpa). As the atomic displacement itself is a function of time for a given neutron flux the dimensions could be represented as a function of time. The conventional unit of measure for irradiation creep and swelling studies is dpa.
  8. Noted – thank you. References will be verified and modified to conform with the journal guidelines.   

Reviewer 3 Report

This is a quality review paper dealing with the mechanical behavior of neutron-irradiated stainless steel. Following are my recommendations for consideration. 

  • Though the review paper deals with mechanical properties and creep, it also deals with swelling in chapter 6. However, the title of the paper doesn't reflect it. I recommend the author modify the title to cover its contents for the interested readers.
  • Chapter 2 has to be modified: The chapter starts discussing phases and compositions, switches gear to talk about He production, and comes back to detail about phases in the steel. It loses the coherency of the chapter. The discussion about He production may be shifted to some other chapters on fracture. 
  • Figure captions have to be checked. Figure 6.8 talks about Figure 15, which is unavailable. Figure 7.7 has no description for figure (b)
  • The creep chapter doesn't deal much with diffusional creep processes such as N-H creep and Coble creep, which are important at low stresses. 

Author Response

Thank you for your insightful comments.

The swelling was included because of the intimate relationship between creep and swelling.  The title will be modified to include swelling and a note has been added in the introduction to indicate how swelling is coupled with creep.

The point on Chapter 2 is well taken.  Chapter 2 has been modified to be more coherent.  I have taken out the text and the Figure on He production and moved to a more appropriate section (5), that on intergranular fracture.

The captions/text related to Figures 6-8 and 7-7 (now figure 7-6) have been corrected – thank you for noticing.  For some reason most of the text was missing for 7-7. It is…… Schematic diagrams showing: (a) the translation of the extra half planes created by the activation of a shear source (S); and (b) hypothetical shear of a prismatic interstitial loop.  Both views are projections perpendicular to the direction of the Burgers vector.  The elastic strains at the surface of the crystal are indicated by the light blue brackets.

In the context of irradiation, diffusional creep is implicit in the linear dependence on stress and the fact that a component of creep is controlled by mass transport (diffusional creep).  Whereas Herring-Nabarro is a high temperature vacancy diffusion creep process, irradiation creep involves both vacancy diffusion and interstitial diffusion, but not necessarily at high temperatures.  For that reason, Herring-Nabarro or Coble creep are not specifically identified as irradiation creep processes. Text has been added to distinguish between high temperature thermal creep based on vacancy diffusion and low temperature irradiation creep that depends on both interstitial and vacancy diffusion.

Reviewer 4 Report

Paper reviews use of stainless steels in nuclear reactors. It is niche content, but clearly presented and would serve readers of Materials.

Paper is very well written and there is really nothing to detract. Some minor mistakes like space or lack thereof before units need to be fixed.

Author Response

Thank you for your kind remarks.  I have reviewed and corrected.

Round 2

Reviewer 1 Report

Author explained why this manuscript was submitted as article not a review paper and it contains useful knowledges and discussions that can benefit readers of this special issue. Therefore, I recommend accepting this manuscript.

Reviewer 2 Report

The peer-reviewed manuscript contains a review of publications devoted to experimental research of the effect of neutron irradiation on the mechanical properties and creep of austenitic stainless steels. The manuscript also discusses the regularities of the formation of localized softening (dislocation channels) in zirconium alloys.

The manuscript also discusses the laws governing the formation of localized softening (dislocation channels) in zirconium alloys. Based on the analysis of the published data, the authors formulated the regularities of swelling of stainless steels under neutron irradiation, changes in their strength characteristics during creep at elevated temperatures and irradiation.

The authors have supplemented and clarified the text of the manuscript. The results of the analysis of experimental and theoretical data are presented more accurately and clearly in the new version of the text of the article.
